# ADAPTIVE PUNISHMENT FOR COOPERATION IN MIXED-MOTIVE GAMES

## ABSTRACT

Mixed-motive scenarios are ubiquitous in real-world multi-agent interactions, where self-interested agents often defect for immediate rewards, overlooking the potential of altruistic cooperation to improve long-term gains and collective welfare. Peer punishment can deter defection, but as costly second-order altruism, its persistent imposition may undermine the punisher's interests. Existing approaches often struggle to effectively implement punishment to promote cooperation. To balance the efficacy and cost of punishment, we propose Adaptive Punishment for Cooperation (APC), a distributed method that determines punishment intensity based on both a dynamic punishment probability and the severity of defection. This dynamic probability substantially reduces costly and ineffective punishment while also promotes cooperation. To accurately assess defection and its severity, we use a defection awareness module, whose learning is guided by game reward. Theoretical analysis and empirical results show APC performs effectively in iterated public goods game. Empirically, APC also significantly outperforms existing baselines across sequential social dilemmas, learning rational and effective punishment policies that foster cooperation by strategically deterring defection.

## 1 INTRODUCTION

Multi-Agent Reinforcement Learning (MARL) provides a framework for training multiple agents to make decisions and interact within a shared environment. Unlike single-agent settings, agents in MARL must account for both environmental dynamics and others' behaviors. Interactions can be cooperative, competitive, or mixed-motive, where cooperation and competition coexist. While MARL has achieved success in cooperative and competitive domains such as StarCraft and SMAC (Lowe et al., 2017; Rashid et al., 2020), mixed-motive games (Leibo et al., 2021) pose greater challenges. The dynamic, uncertain agent relationships make it difficult for agents to form stable expectations or trust, often exacerbating the tendency to prioritize short-term personal gains over long-term collective welfare. A key challenge is training agents to avoid social dilemmas like defection and foster cooperation for mutually beneficial outcomes.

A commonly used training paradigm in MARL is Centralized Training with Decentralized Execution (CTDE), which assumes agents share information and jointly optimize collective rewards (Sunehag et al., 2017). However, this assumption is often unrealistic in social dilemma settings where agents are self-interested. In contrast, Decentralized Training with Decentralized Execution (DTDE) can lead agents to converge to suboptimal local equilibria, failing to escape social dilemmas (see Figure 3). Consequently, even reward-maximizing agents may not achieve desired collective outcomes. These limitations reveal the need for guiding decentralized agents out of suboptimal equilibria and converging toward more socially beneficial outcomes in mixed-motive scenarios.

To promote cooperation in mixed-motive games, it is crucial to both deter defection and incentivize collective welfare. Inspired by human societies, punishment raises the cost of defection, discouraging short-term selfish behavior and guiding agents toward long-term group benefits (Henrich et al., 2006). However, peer punishment imposes a penalty on others while incurring a cost for the punisher (e.g., time and effort spent stopping someone from smoking in public). That is, punishment is second-order altruism. Excessive punishment may harm the punisher's own interest, and agents may avoid punishing due to its cost, failing to achieve cooperation (Köster et al., 2022). To address these challenges, we propose a distributed decision-making method named APC (Adaptive Punishment

for Cooperation), which is context-sensitive and adaptively adjusts the probability and severity of punishment. Guided by reward signals, APC learns a defection awareness module to evaluate the defectiveness of others' actions, which determines the probability and intensity of punishment. Notably, to prevent ineffective punishment and avoidable cost, punishment probability is dynamically adjusted according to the historical effectiveness of punishment, based on the reduction in defections.

The main contributions of this work are: **(1)** We propose a novel opponent-adaptive punishment method, APC, that dynamically adjusts punishment intensity to avoid ineffective punishment and excessive costs, thereby promoting cooperation in mixed-motive games. **(2)** We develop a self-learning defection awareness algorithm for detecting different degrees of defection in multi-agent interactions. **(3)** We provide theoretical analysis and demonstrate empirically that APC outperforms existing baselines across Iterated Public Goods Game and Sequential Social Dilemmas (SSDs).

## 2 Preliminaries

### 2.1 Partially Observable Markov Game

We consider an $N$-player Partially Observable Markov Game (POMG) (Littman, 1994; Oliehoek, 2012), defined as:
$$M = \langle N, \mathcal{S}, \{\mathcal{O}^i\}, \{\mathcal{A}^i\}, T, \{R^i\} \rangle,$$
where $N$ is the number of agents, $\mathcal{S}$ is the state space of the environment, $\mathcal{O}^i$ denotes the observation space of agent $i$, and $\mathcal{A}^i$ represents the action space of agent $i$. $s \in \mathcal{S}$ represents the state of the environment. The state transition function $T : \mathcal{S} \times \mathcal{A}^1 \times \cdots \times \mathcal{A}^N \times \mathcal{S} \to [0, 1]$ defines the probability distribution over the set $\mathcal{S}$, representing the probability of transitioning to state $s'$ given the current state $s$ and a joint action $\vec{a} = (a^1, \ldots, a^N) \in \mathcal{A}^1 \times \cdots \times \mathcal{A}^N$.

Each agent receives a local observation $o^i$ and selects an action according to its policy $\pi^i(a^i|o^i)$. The environment returns a reward $r^i = R^i(s, \vec{a})$ to agent $i$. The objective of each agent is to maximize its expected long-term return, which is defined as

$$V_i^{\pi^i}(s_0) = \mathbb{E}_{\vec{a}_t \sim \pi, \, s_{t+1} \sim T(s_t, \vec{a}_t)} \left[ \sum_{t=0}^{\infty} \gamma^t r^i \right], \tag{1}$$

where $\gamma \in [0, 1)$ is the discount factor.

### 2.2 Policy Gradient Learning

In decentralized MARL, each agent $i$ aims to maximize its own expected return, as defined in Eq.(1). We use policy-based Advantage Actor-Critic (A2C) method (Sutton et al., 1999; Mnih et al., 2016) as agents' learning algorithm. The policy $\pi^i$ for agent $i$ is parameterized by $\theta^i$. The critic evaluates actor using TD-error computed as $R^i(s_t, \vec{a}_t) + \gamma V^{\pi_{\theta^i}}(\vec{o}_{t+1}) - V^{\pi_{\theta^i}}(\vec{o}_t)$, where $\vec{o}_t = (o_t^1, \ldots, o_t^N)$ denotes the joint observation of all agents at timestep $t$.

## 3 Methodology

To address the dilemma of cooperation and defection in multi-agent systems, we propose a distributed MARL method, named Adaptive Punishment for Cooperation (APC). As shown in Figure 1, APC consists of two core modules: defection awareness and adaptive punishment. The defection awareness module takes agents' local observation trajectories as input and produces a probability distribution over the opponent's actions, where actions with higher probabilities are regarded as more severe defections. The adaptive punishment module then uses both the defection judgment and the variation in defection frequency as input to generate a dynamic punishment intensity—that is, the more frequent the defections, the harsher the punishment. Moreover, it adjusts the punishment probability based on the effectiveness of punishment in reducing defections. Next, we take agent $i$ as the focal agent.

### 3.1 Defection Awareness

We propose a defection awareness method to train the Defection Predictor Network $\mu^i$ to identify defection, where the network learns to predict the behavior of a target agent $j$ that would harm agent

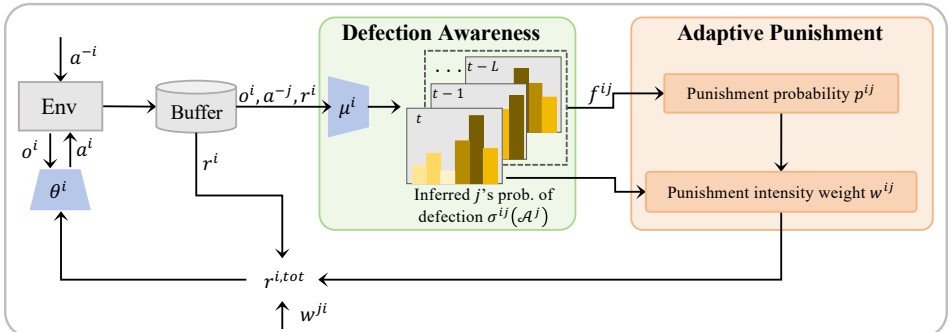

Figure 1: **Overview of the APC Framework.** The framework consists of two components: Defection Awareness and Adaptive Punishment. Any Agent $i$ interacts with the environment to collect trajectories, stored in a replay buffer to train the defection predictor network $\mu^i$. $\mu^i$ is trained with the defection awareness method, using $o^i$, $a^{-j}$ and $r^i$ to predict the probability distribution of defection $\sigma^{ij} = p_{\mu^i}(\cdot|o^i, a^{-j})$ over $\mathcal{A}^j$. A higher-probability action $a^j$ indicates a greater degree of defection. Adaptive punishment operates as follows. First, the punishment probability $p^{ij}$ is computed using $f^{ij}$ (Eq.(3)), which reflects the defection frequency of agent $j$ over the past $L$ timesteps. Next, the punishment intensity weight $w^{ij}$ is determined: if $\sigma^{ij}(a^j) > \frac{1}{|\mathcal{A}^j|}$, $w^{ij}$ is set to be proportional to $\sigma^{ij}(a^j)$ with probability $p^{ij}$; otherwise, $w^{ij} = 0$. Finally, agent $i$'s total reward for updating policy network $\theta^i$ is $r^{i,\text{tot}} = r^i - \sum_{j=1, j\neq i}^{N} w^{ij}c - \sum_{j=1, j\neq i}^{N} w^{ji}\delta$, where $r^i$ is the original reward without punishment, and $c$, $\delta$ are the unit cost and penalty. APC 's pseudocode is given as Algorithm 1.

$i$'s reward $r_t^i$ more. $\mu^i$ outputs probability distribution of defection $\sigma_t^{ij} = p_{\mu^i}(\cdot|o_t^i, a_t^{-j})$ over the agent $j$'s action space $\mathcal{A}^j$, given agent $i$'s observation $o_t^i$ and the joint actions of all other non-target agents $a_t^{-j}$. The dimensionality of $a_t^{-j}$ is fixed, and actions of agents outside of the observation range are represented using a default placeholder value of $-1$. If $\sigma_t^{ij}(a_t^j) > \frac{1}{|\mathcal{A}^j|}$, $a_t^j$ is considered a defection by agent $i$. When the probability of $a_t^j$ exceeds the mean, it indicates that this action being unfavorable for agent $i$'s payoff relative to the average payoff from other actions for agent $i$ which is proven in appendix C. The training objective of $\mu^i$ is formalized as a maximization problem:

$$J(\mu^i) = \mathbb{E}_{a_t^j \sim p_{\mu^i}} \left[ -r_t^i(s_t, \vec{a}_t) + \beta H(\sigma_t^{ij}) \right]. \tag{2}$$

Here, $-r_t^i(s_t, \vec{a}_t)$ drives the network to minimize the reward, and $\beta H(\sigma_t^{ij})$ incorporates entropy regularization to encourage behavioral diversity and prevent premature convergence to a peaked distribution. $H(\sigma_t^{ij}) = \mathbb{E}_{a_t^j \sim \sigma_t^{ij}} \left[ -\log \sigma_t^{ij}(a_t^j) \right]$ denotes the entropy of the distribution $\sigma_t^{ij}$, which measures its randomness: a larger entropy indicates a more uniform distribution. The regularization parameter $\beta$ the entropy's importance in the optimization objective. The objective function $J(\mu^i)$ is designed to guide agent $i$ toward learning to identify defection with sufficient exploration.

Through this process, $\mu^i$ effectively identifies the behaviors of target agent $j$ that would more harm agent $i$'s reward, given different $o_t^i$ and $a_t^{-j}$. Once $\mu^i$ has converged, its parameters are fixed during the training of policy network $\theta^i$, ensuring stable defection awareness. This learned policy serves as a criterion for identifying defection actions in an opponent-aware manner. The entire training is performed via gradient descent using multi-agent trajectory data.

## 3.2 ADAPTIVE PUNISHMENT

We introduce an adaptive punishment mechanism to guide agents toward rational and graded punitive behavior. Adaptivity is reflected in two aspects: (1) punishment probability is dynamically adjusted according to whether past punishments have successfully reduced defections; and (2) the punishment intensity is modulated based on the degree of defection.

### 3.2.1 PUNISHMENT PROBABILITY

The punishment probability $p_t^{ij}$ denotes the probability with which agent $i$ punishes agent $j$ upon defection. $p_t^{ij}$ is dynamically adjusted to prevent ineffective punishment, in which past punishments

have failed to reduce defections. It is evaluated by the change of defection frequency of agent $j$ from the perspective of agent $i$, denoted as $f_t^{ij}$, where defection is determined by $\mu_i$. Specifically, $f_t^{ij,s}$ denotes the value of $f_t^{ij}$ in window $s$, where each window spans $L$ timesteps. If $f_t^{ij,s}$ does not decrease compared to $f_t^{ij,s-1}$, the punishment in window $s$ is deemed ineffective. Consequently, $p_t^{ij}$ should be reduced in window $s+1$. We compute $p_t^{ij}$ in window $m$ as follows:

$$p_t^{ij} = 1 - \frac{\sum_{s=1}^{m-1} \mathbf{1}\left[\left(f_t^{ij,s} \geq f_t^{ij,s-1} \vee \left|f_t^{ij,s} - \frac{1}{s}\sum_{k=1}^{s} f_t^{ij,s-k}\right| < \varepsilon\right) \wedge f_t^{ij,s} \geq \varepsilon\right]}{m-1}. \quad (3)$$

Here, $\mathbf{1}[\cdot]$ is the indicator function that outputs 1 if the condition inside is true, indicating that the punishment in window $s$ is considered ineffective, and 0 otherwise and $\varepsilon$ is a tolerance threshold. The condition $f_t^{ij,s} \geq f_t^{ij,s-1}$ indicates that the punishment in window $s$ failed to reduce $f_t^{ij}$. The term $\left|f_t^{ij,s} - \frac{1}{s}\sum_{k=1}^{s} f_t^{ij,s-k}\right| < \varepsilon$ suggests that $f_t^{ij}$ in window $s$ has not significantly decreased compared to the average defection frequency over the past $s-1$ windows. The constraint $f_t^{ij,s} \geq \varepsilon$ avoids misinterpreting a low defection frequency as signs of ineffectiveness. When $f_t^{ij,s}$ is very small, the punishment in window $s$ is still effective because it helps maintain cooperation. These suggest that agent $j$ is unlikely to change its behavior in response to punishment, and continuing to apply punishment in such cases would lead to unnecessary costs and a decrease in $r_t^{i,\text{tot}}$. Specifically, punishment is considered effective by default in window 0 and 1. Accordingly, $p_t^{ij}$ is initialized to 1 in window 0 and 1, and is subsequently updated after window 1 based on Eq.(3).

### 3.2.2 Punishment Intensity

Any agent $i$ holds a Punishment Intensity Weight (PIW) vector at time step $t$: $w_t^i = \left[w_t^{ij}\right]_{j=1}^N$, where $w_t^{ij} \in [0,1]$ is the fraction of agent $i$'s punishment to agent $j$. Let $B_t^{ij}$ be a Bernoulli random variable with success probability $p_t^{ij}$, i.e., $B_t^{ij} \sim \text{Bernoulli}(p_t^{ij})$. $w_t^{ij}$ is computed as follows:

$$w_t^{ij} = B_t^{ij} \cdot \begin{cases} 0, & \sigma_t^{ij}(a_t^j) \leq \frac{1}{|\mathcal{A}^j|}, \\ \dfrac{\sigma_t^{ij}(a_t^j)}{\max\limits_{a \in \mathcal{A}^j} \sigma_t^{ij}(a)}, & \sigma_t^{ij}(a_t^j) > \frac{1}{|\mathcal{A}^j|}. \end{cases} \quad (4)$$

If agent $i$ punish agent $j$, agent $i$ pays $w_t^{ij}c$ and agent $j$ loses $w_t^{ij}\delta$. In partially observable environments, agent $i$ can only punish $j$ if $j$ is within its observation range; otherwise, punishment is not applied. Agent $i$'s total reward is $r_t^{i,\text{tot}} = r_t^i - \sum_{j=1,j\neq i}^N w_t^{ij}c - \sum_{j=1,j\neq i}^N w_t^{ji}\delta$, where $r_t^i$ is the original reward from environment without punishment. Policy $\pi_{\theta^i}^i(a_t^i|o_t^i)$ is trained to maximize $\mathbb{E}_{\pi^i}\left[\sum_{t=0}^H \gamma^t r_t^{i,\text{tot}}\right]$ using the TD-error (see Section 2.2), where $H$ denotes the length of an episode.

## 4 Experiment

### 4.1 Experiment Setup

#### 4.1.1 Environments

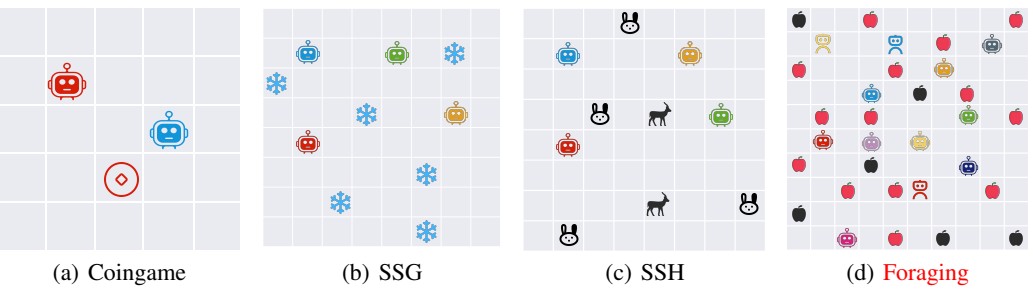

|     (a) Coingame     |     (b) SSG     |     (c) SSH     |     (d) Foraging     |

Figure 2: Graphic representations of four SSDs: (a) Coingame (5×5 map), 2 agents, (b) SSG (8×8 map), 4 agents, (c) SSH (8×8 map), 4 agents, (d) Foraging (10×10 map), 12 agents.

We evaluate APC on both classical and spatiotemporal-extended SSDs mixed-motive games, illustrated in Figure 2. The advantages of APC over traditional methods (such as Tit-for-Tat) in classical mixed-motive game can be found in Appendix A. Below are detailed descriptions of the characteristics and design principles for all environments.

**Iterated Public Goods Game (IPGG).** Iterated Public Goods Game extends the two-player game in Iterated Prisoner's Dilemma (IPD) (Foerster et al., 2018) to a multi-player game, better modeling the evolution of group cooperation. In each round, $n$ players independently decide whether to contribute their endowment $e$ to a common pool. The total contributions are multiplied by $r$ ($1 < r < n$) and evenly distributed among all players, regardless of contribution. Non-contributors keep their endowment, making defection individually advantageous. We use parameters $[n, e, r] = [5, 1, 3]$.

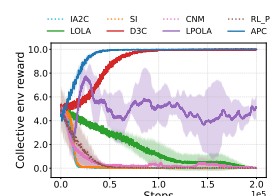

Figure 3: Learning curves for IPGG of self-play training. The curve plotting rules and meanings are consistent with those in Figure 4.

**Coingame**. Coingame (Lerer & Peysakhovich, 2017) is a spatial and temporal extension of IPD. One agent is assigned the color Red, and the other agent is assigned the color Blue. The environment contains red and blue coins. An agent receives 1 point for picking up any coin. However, if an agent picks up a coin matching the other agent's assigned color, the other agent loses 2 points.

**Sequential Snowdrift Game (SSG).** In SSG, there are 6 snow piles that need to be cleared. Once a snow pile is cleared, all agents receive 6 points, while the clearing agent incurs a cost of 4 points. Agents have a choice between cooperating to clear snow or free-riding.

**Sequential Stag-Hunt (SSH).** This environment is inspired by Markov Stag Hunt in (Peysakhovich & Lerer, 2018) and used in (Kong et al., 2024). In SSH, four agents choose between hunting a hare individually or cooperating to hunt a stag. Hunting a hare is easier but yields 1 point, while hunting a stag requires cooperation (at least 2 agents) and gains 10 points, shared equally among participants.

**Foraging**. The design of this environment is based on (Köster et al., 2022). In a 2D foraging game, 12 agents gather berries. 75% of them are common agents that can only collect common berries (3 points); 25% are special agents that can also collect forbidden berries (4 points). However, once a special agent collects a forbidden berry, it triggers permanent resource degradation, reducing the reward of all common berries to 1 point and negatively affecting the rewards of all common agents.

### 4.1.2 IMPLEMENTATIONS

We adopt the DTDE architecture, with each agent having independently parameterized networks. The defection predictor network consists of two convolutional layers for observation encoding, an LSTM layer for temporal modeling, and several ReLU-activated fully connected layers. The input is a multi-channel binary tensor, with channel count varying by environment. The policy network follows an actor-critic architecture, where both actor and critic share a similar architecture to the defection predictor, including convolutional, LSTM, and fully connected layers. The actor's output dimension matches that of the defection predictor, while the critic's output is a single scalar value.

For IPGG, which lacks spatial structure, we simplify the architecture by removing convolutional and LSTM layers, reducing parameter counts in fully connected layers, and increasing the learning rate to improve training efficiency. Additional implementation details are provided in Appendix E.

### 4.1.3 BASELINES

Independent Advantage Actor-Critic (IA2C) is a classic gradient-based reinforcement learning algorithm suitable for agents to learn policies under completely independent conditions (Mnih et al., 2016). LOLA considers the learning process of other agents when updating its own policy parameters (Foerster et al., 2018). SI achieves coordination by rewarding agents for having causal influence over other agents' actions (Jaques et al., 2019). D3C guides self-interested agents toward collectively efficient cooperative equilibria by having them mix rewards and follow the gradient of an efficiency bound during learning (Gemp et al., 2022b). To enable punitive capability, all above methods are equipped with a punitive action within their action spaces. We additionally introduce three punishment-based methods for comparison. LPOLA simultaneously predicts environmental

actions and determines which actions of other agents to penalize, applying penalties when predictions match reality (Schmid et al., 2021). CNM fosters cooperation by establishing social norms, whose punishments are socially enforced based on group consensus (Yaman et al., 2022). The core idea of both LPOLA and CNM methods is to encourage punishment by attaching pseudo-rewards. RL Punish utilizes two distinct networks: a policy network for making primary action decisions and a punishment network for deciding which agents to penalize. Both networks are trained via the Advantage Actor-Critic (A2C) algorithm with the objective of maximizing the collective environmental reward.

## 4.2 MAIN RESULTS

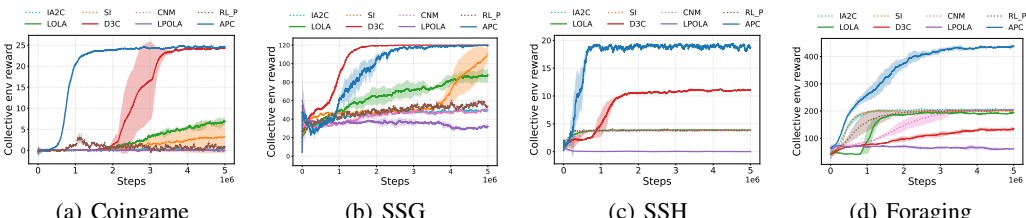

| (a) Coingame | (b) SSG | (c) SSH | (d) Foraging |

Figure 4: Learning curves for four types of SSDs of self-play training. The curves represent the collective environment rewards, with all data collected from five training runs using different random seeds. The solid lines indicate the mean across runs, and the shaded areas represent the standard deviation. Note that all experimental results in this paper are evaluated over five random seeds, and this information will be omitted in the subsequent figure captions for brevity.

In IPGG, APC and D3C successfully avoid inefficient Nash equilibria, leading both agents to achieve nearly 100% cooperation (Figure 3). APC's result aligns with our theoretical analysis (shown in Appendix B.1) and demonstrates APC's significant advantage in collective rewards, reaching the best reward level. While LPOLA can achieve a certain degree of cooperation, some individuals within the group still choose to defect, preventing the attainment of optimal benefits. In addition, other baseline methods fail to overcome the dilemma of defection; their cooperation rates rapidly decline to low levels, and the final performance differences among them are minimal. The result demonstrates that APC performs excellently in the classical multi-player game scenarios, exhibiting strong ability in fostering cooperation.

In Coingame, APC most rapidly guides agents to avoid collecting coins of other agents' colors, significantly improving collective rewards (Figure 4(a)), whereas D3C achieves this at a relatively slower pace. Although LOLA and SI learn to some extent to prevent agents from picking other agents' coins, both fail to completely avoid such behavior. For CNM, both agents eventually converge on punishing all actions, which leads to solely optimizing pseudo-rewards and results in very low total returns—as can be observed in Figure 13(a). Consequently, none of the agents successfully learn to collect coins. In addition, agents of other baselines fail to learn an effective punishment strategy and fail to realize that picking coins of other agent's color would lead to punishment. As a result, they continue to selfishly collect coins.

In SSG, APC agents learn to clear all snow piles faster than D3C, SI, and LOLA, achieving near-optimal collective rewards (Figure 4(b)). However, LPOLA, due to interference from pseudo-rewards associated with punitive actions, neglects the pursuit of actual environmental rewards, falling below IA2C. Other baselines remain trapped in the snowdrift dilemma—expecting others to clear snow while free-riding—leaving numerous piles uncleared.

In SSH, agents in APC, concerned about being punished by other agents for hunting hares, ultimately choose to cooperate in hunting stags, thereby maximizing collective rewards (Figure 4(c)). D3C gets it into the lazy problem (Sunehag et al., 2017), and early hunting leads to moving out of the environment and failing to obtain the group rewards of others' later hunting. Thus, D3C may not hunt until the last few steps, and likely miss the the opportunity of cooperating to hunt stags. Due to pseudo-rewards, LPOLA also neglects the pursuit of actual environmental rewards, resulting in performance inferior to IA2C. In other baseline methods, agents, seeing only short-term personal rewards, choose to hunt hares, failing to engage in group cooperation. Moreover, across different initial strategies (e.g., some starting with hare-hunting), the overall performance and collective rewards in SSH (Table 3) stay largely consistent, clearly showing our method's robustness to initialization.

In Foraging, APC successfully prevents special agents from collecting forbidden berries, thereby enhancing the subsequent collective returns and achieving optimal performance (as shown in Figure 4(d)). In contrast, although D3C performed well in IPGG and the first three SSDs, its performance in Foraging was even worse than that of IA2C. Meanwhile, LPOLA was still affected by pseudo-rewards, which led to its performance being inferior to IA2C. Other baseline methods resulted in common agents consistently collecting common berries while special agents persistently pursued forbidden berries, with none learning to prevent the special agents from collecting the forbidden ones. This failure led to a significant gap between the overall reward and the optimal outcome.

It is worth noting that due to the significant influence of the punishment on RL Punish, its collective rewards values are significantly lower than those of other methods. Due to the scaling of the reward axis, the superiority of APC is not fully displayed. Therefore, in the collective environment rewards in the main text, we have not considered the impact of $c$ and $\delta$. Of course, if the effect of the punishment on rewards is taken into account, APC still demonstrates optimal results, with specific analysis available in the experimental results in Figure 13.

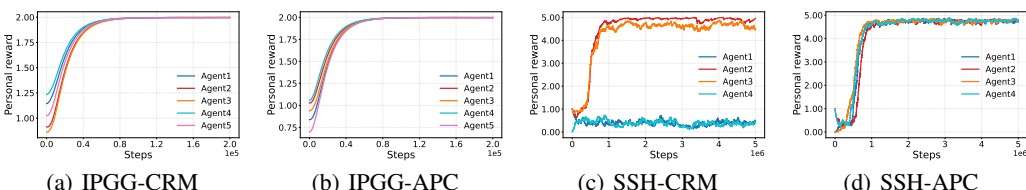

| (a) IPGG-CRM | (b) IPGG-APC | (c) SSH-CRM | (d) SSH-APC |

Figure 5: The learning curves of each agent in IPGG and SSH for CRM (Collective Reward Maximization) and APC. CRM aims to maximize collective reward. To highlight each agent's behavior, results from a random seed are shown, since different seeds lead to different task divisions.

In addition to collective rewards, we also analyzed the **fairness** of APC at the individual reward level in both IPGG and SSH. Fairness is a key metric for evaluating multi-agent systems, as it ensures balanced resource allocation and maintains agents' motivation. In real-world scenarios, fair systems are more socially acceptable and help reduce conflict. Figure 5 presents the individual reward training curves for the CRM (Collective Reward Maximization) method—which aims to maximize collective rewards—and for APC. We evaluate fairness using the metric $Equality(E)$, defined as $E = 1 - G_P$, where $G_P = G \cdot \frac{T_a - T_n}{T_a + T_n} \in [0, 1]$ (De Battisti et al., 2019). Here, $G = \frac{\sum_{i=1}^{N} \sum_{j=1}^{N} |R_i - R_j|}{2N \sum_{i=1}^{N} R_i}$ is the Gini index, $T_a = \sum_{R_i \geq 0} R_i$ denotes the total positive rewards, and $T_n = \sum_{R_i < 0} |R_i|$ denotes the total magnitude of negative rewards. A higher value of $E$ indicates greater fairness. The fairness results are averaged over five random seeds, and the individual reward curves for the other seeds can be found in Figure 14 and 15. In IPGG, both CRM and APC achieve fairness values close to 1. However, the results in SSH show that $E(\text{CRM}) \approx 0.598$, while $E(\text{APC}) \approx 0.987$, demonstrating that APC significantly improves fairness compared to the SOTA CRM algorithm.

### 4.3 ABLATION STUDY

To further evaluate the performance of Defection Predictor Network (DPN) and Adaptive Probability (APr) (Eq.(3)), two ablation experiments are designed. The first ablation experiment is as follows: we removed the trained DPN and replaced it with a randomly initialized policy network to verify the critical role of a reasonable defection predictor in improving overall performance. This method is referred to as APC w/o DPN. The second ablation experiment is as follows: rather than using an adaptive probability, we simply set it to 1 throughout, which is denoted as APC w/o APr.

The results of two ablation experiments are shown in Figures 11-12 and Figures 16-17. For APC w/o DPN, due to the failure to learn effective and reasonable punishment policy, it is unable to leverage the advantages of the punishment. For environments in Figure 2, the performance of APC w/o DPN was generally poor, there was no significant improvement in performance. For APC w/o APr, in the self-training setting (i.e., APC w/o APr agents interacting with each other), opponents learn to avoid punishment and reduce their defection behaviors, leading to low punishment frequency, despite the agent maintaining a high probability $p^{ij}$. As a result, its performance remains close to that of APC. However, when facing rule-based opponents (see Figure 6(a) and 6(c)), APC w/o APr agent sustains high punishment frequency, failing to adapt. This highlights the importance of adaptive

probability in avoiding ineffective and excessive punishment when opponents do not respond to it. It is worth noting that in the results of APC w/o APr vs Defect in Figure 6(c), the noise is caused by the stochastic nature of SSH, where the number of hare-hunting actions (defections) may vary across different time windows.

## 4.4 ADAPTIVE PUNISHMENT CAPABILITY

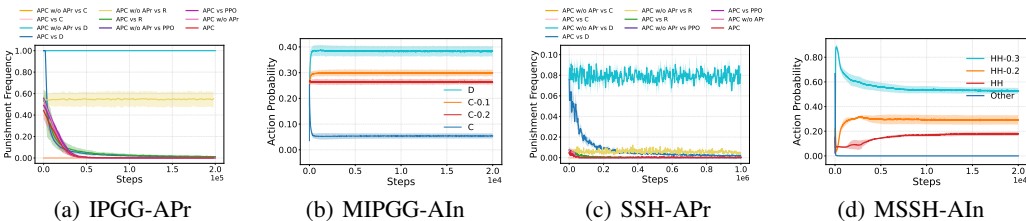

(a) IPGG-APr    (b) MIPGG-AIn    (c) SSH-APr    (d) MSSH-AIn

Figure 6: Adaptive punishment capability of APC. (a,c) APC adjusts punishment frequency: high for defection, low for cooperation, and decreases when punishment proves ineffective. (b,d) Learning curves of the action probability distribution of DPN in MIPGG and MSSH. APC scales Adaptive Intensity (AIn): stronger for severer defections, achieving proportional response.

We evaluate the adaptive punishment capability of APC from two perspectives. First, **adaptive punishment frequency**: we examine whether APC can dynamically adjust its punishment frequency based on opponents' behavioral responses to protect its long-term interests. Second, **adaptive punishment intensity**: we examine whether APC can adjust its punishment strength when confronted with defections of varying severity, such that severer defections receive stronger sanctions.

**Adaptive Punishment Frequency.** We design the following setup: a focal APC agent interacts with three types of rule-based agents and other shapeable opponents (e.g., PPO, APC). The rule-based agents include always-defecting, always-cooperating, and randomly acting agents. In IPGG, defectors never contribute, while cooperators always contribute; in SSH, defectors always hunt hares, whereas cooperators hunt stag. As shown in Figure 6(a) and 6(c), the focal APC agent initially applies much higher punishment frequency to defectors and random agents, while keeping it near zero for cooperators, showing ability to detect defection. As training progresses, punishment behavior self-regulates: when punishment has no effect, the focal agent reduces probability (Eq.(3)) to avoid excessive punishment; if it successfully induces PPO or APC agents to reduce defection, punishment frequency also declines. Moreover, the collective rewards of APC vs PPO in both IPGG and SSH, in Figure 18(a) and 18(b), are close to the results shown in Figure 3 and 4(c). These results show that APC dynamically adjusts punishment frequency while constraining defection.

**Adaptive Punishment Intensity.** We modify the environments as follows: in Modified IPGG (MIPGG), we introduce two partial-contribution actions, contributing 0.2 (C-0.2) and 0.1 (C-0.1). Compared with full contribution (C), C-0.2, C-0.1, and no contribution (D) harm the focal APC agent's interests and represent different levels of defection. In Modified SSH (MSSH), we add two actions: hunting hare while reducing others' payoffs by 0.2 (HH-0.2) or 0.3 (HH-0.3). Compared with other actions (e.g., moving or hunting stag, collectively denoted as Other), Hunt Hare (HH), HH-0.2, and HH-0.3 are detrimental to the focal agent. We train DPN to assess its ability to detect defections of different severities and impose proportionate punishments. As shown in Figure 6(b), in MIPGG, DPN assigns PIW of [1, 0.79, 0.70, 0] to [D, C-0.1, C-0.2, C]. In Figure 6(d) of MSSH, PIW for [HH-0.3, HH-0.2, HH, Other] are [1, 0.543, 0.360, 0], following Eq.(4). These results show APC flexibly adjusts PIW to match defection severity, achieving proportional response.

## 4.5 HYPERPARAMETER SENSITIVITY

The hyperparameter analysis includes $(c, \delta)$, $\beta$, and $(\varepsilon, L)$. To examine the impact of parameters $c$ and $\delta$ on the cooperation rate in IPGG, we conduct experiments where both parameters vary from 0 to 1.4 in increments of 0.1. Additionally, to evaluate the effect of $\beta$, we conduct three experiments in SSH and MSSH with different $\beta$ settings: 0.001, 0.005, and 0.01, to capture $\beta$'s influence on PIW. To investigate the impact of $\varepsilon$ and $L$ on punishment frequency, we show training curves of collective rewards for APC agents under self-play, as well as changes in punishment frequency when interacting

with always-defecting and random agents in IPGG. The experiments use the following $(\varepsilon, L)$ settings: (0.005, 2000), (0.01, 1000), (0.01, 2000), (0.01, 3000), and (0.015, 2000). $\varepsilon$ is selected to be as small as possible while remaining robust to occasional defections due to randomness. $L$ is chosen to be $\geq$ the number of training timesteps required for parameter update of the policy network.

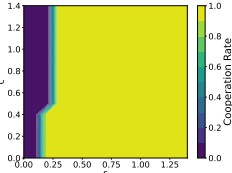

As shown in Figure 7, when $\delta \geq 0.3$, $c \in [0, 1.4]$ and $\delta = 0.2$, $c \in [0, 0.4]$, the cooperation rate converges to full cooperation in IPGG. However, when $\delta = 0.2$ and $c \in [0.5, 1.4]$, full cooperation is not achieved. This is because a small $\delta$ exerts insufficient deterrence on defectors, while a large $c$ hinders the learning of effective cooperative policies. Once $\delta$ is slightly increased, the negative impact of $c$ diminishes, and cooperation emerges robustly. These results indicate that the cooperation rate is generally robust to variations in $c$ and $\delta$, as long as $\delta$ is sufficiently large to suppress defection. Importantly, our theoretical analysis reveals a key result: under the intervention of the APC method, the expected reward for an agent choosing to defect is given by $-4\delta$, while the reward for choosing to cooperate is $-0.4$. This means that when $\delta > 0.1$, cooperation becomes the rational choice, which directly explains the cooperative patterns observed in Figure 7. Note when $\delta$ is set to 0.1 or 0.2 and $c$ exceeds 0.4, the cooperation rate remains low. This is likely because a high value of $c$ interrupts the learning process when $\delta$ is insufficient, weakening the incentive effect of punishment. A more detailed theoretical analysis is provided in Appendix B.2.

Figure 7: Cooperation rate of APC after convergence across different combinations of $c$ and $\delta$ in IPGG. The X-axis and Y-axis represent the values of $\delta$ and $c$, respectively, where $\delta, c \in [0 : 0.1 : 1.4]$.

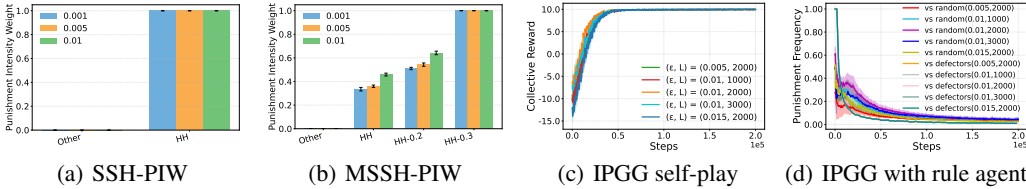

| (a) SSH-PIW | (b) MSSH-PIW | (c) IPGG self-play | (d) IPGG with rule agents |

Figure 8: Hyperparameter sensitivity results on $\beta$ and $(\varepsilon, L)$. (a)(b) show DPN's PIW in SSH and MSSH. (c) shows APC reducing defection in self-play. (d) shows APC lowering punishment when opponents keep defecting, indicating robustness.

As shown in Figure 8(a), in SSH, PIW assigned by DPN to the defection behavior HH remains consistently 1 across all three $\beta$ configurations, while PIW for Other behaviors is always 0. As shown in Figure 8(b), in MSSH, the PIW assigned by DPN to the three levels of defection behaviors (HH, HH-0.2, HH-0.3) are approximately the same, and remain 0 for Other behaviors. As the $\beta$ increases, more policy randomness is taken into account, which results in slightly higher PIW for HH and HH-0.2. Moreover, under these three $\beta$ configurations, the collective rewards in MSSH (Figure 18(c) and 18(d)) are close to 20, which is consistent with the results shown in Figure 4(c). These findings indicate that the APC method exhibits a certain degree of robustness with respect to $\beta$.

As shown in Figure 8(c), during self-play training, APC agents are able to punish defectors across different $(\varepsilon, L)$ settings, which effectively reduces defection frequency and promotes cooperative behavior among agents. Furthermore, when the focal APC agent interacting with random or always-defecting opponents, as shown in Figure 8(d), the focal APC agent can recognize that its punishment has no effect on the opponent's behavior and subsequently reduces its punishment frequency over time, eventually approaching zero. This demonstrates the robustness of APC under different $(\varepsilon, L)$.

## 5 RELATED WORK

Multi-agent reinforcement learning has been widely applied to address problems related to social dilemmas. Various reward mechanisms, including centralized redistribution (Gemp et al., 2022a) and decentralized peer-to-peer rewarding (Kong et al., 2024), have been proposed to promote cooperation by aligning individual and collective interests in mixed-motive games. Punishment, which promote cooperation without requiring additional reward resources, have gained increasing attention. In evolutionary game theory, punishment is widely regarded as a key factor in achieving stable cooperation (Sun et al., 2023; Salahshour, 2021).Some approaches (Li et al., 2021; Wang

et al., 2021; Flores et al., 2021) rely on centralized controllers or self-organized structures among individuals to implement punishment, further investigating how punishment influence the evolution and stability of cooperative behaviors. Other methods (Gao et al., 2020; Song et al., 2020; Ohdaira, 2022) adopt conditional punishment policy, where the decision to punish depends on conditions such as the difference between an individual's own payoff and the average payoff of its neighbors, thereby improving the specificity and effectiveness of punishment.

In MARL, although some studies have attempted to incorporate punishment method into multi-agent systems, some designs remain relatively simple—typically adding a punishment action into the action space (Leibo et al., 2021; Hughes et al., 2018; Pérolat et al., 2017). However, experimental results, shown as in Figure 4, have shown that merely relying on such punishment action combined with standard Independent MARL methods often fails to promote cooperation in SSDs. Other approaches (Wang et al., 2023; Dasgupta & Musolesi, 2025) depend on auxiliary mechanisms or specific environmental assumptions to support the effectiveness of punishment, such as introducing reputation systems or assuming that punishment behaviors can generate positive incentives. Some methods further assume that punishment can directly yield positive rewards for the punisher (Schmid et al., 2021; Yaman et al., 2022), which is clearly unrealistic. We propose APC, which does not rely on environmental assumptions but instead identifies defection behaviors through a defection predictor network and applies targeted punishment. This guides agents to learn reasonable and effective punishment policy, thereby promoting cooperation more effectively.

Existing opponent modeling approaches (Wang et al., 2020; Lu et al., 2022) mainly focus on learning dynamics or modeling the distribution of opponent policies. For example, LOLA (Foerster et al., 2018) explicitly models the opponent's learning process through gradient backpropagation, while SOM (Raileanu et al., 2018) attempts to infer opponent behavior from the agent's own perspective. From the perspective of opponent defection, we propose an explicit defection recognition method, leveraging a defection awareness module, enabling agents to identify different degrees of defection.

## 6 CONCLUSIONS

We propose Adaptive Punishment for Cooperation (APC), which combines dynamic punishment intensity with explicit defection awareness to align punishment intensity with defection severity, reducing exploitation and defection frequency. A limitation of this work is that APC has only been evaluated in relatively simplified environments. Its effectiveness and scalability in more complex and realistic multi-agent scenarios remain to be validated. Additionally, this work focuses solely on punishment; how APC complements reward-based method for cooperation remains an open question.

ETHICS STATEMENT

This work focuses on developing adaptive punishment mechanisms to promote cooperation in mixed-motive multi-agent environments. All experiments were conducted in simulated environments without the use of human subjects, personal data, or sensitive information. The proposed method is intended for advancing research in multi-agent reinforcement learning and the study of cooperation in artificial systems. This work may have the potential to positively impact society by fostering collaboration in distributed AI systems.

REPRODUCIBILITY STATEMENT

We provide comprehensive details to ensure the reproducibility of our work. The model architectures, optimization algorithms, and hyperparameters are described in Section 4.1.2 and Appendix F. The experimental environments, including rules, state/action spaces, and reward structures, are thoroughly specified in Section 4.1.1 and Appendix E. To account for randomness, all reported results are averaged over five independent runs with different random seeds, and the corresponding standard deviations are provided. Implementation details such as network initialization strategies, learning rates, and batch sizes are explicitly stated. The hardware setup used for training (NVIDIA GPUs with PyTorch) is also documented. In addition, we plan to release the full source code, along with configuration files and scripts for reproducing all experiments, upon publication.

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

## A  ANALYSIS IN ITERATED PRISONER'S DILEMMA

Iterated Prisoner's Dilemma (IPD) is a widely used two-agent bench-
mark for comparability with prior work Foerster et al. (2018); Lerer
& Peysakhovich (2017); Yang et al. (2020). IPGG is a multi-agent
extension of IPD. Before introducing the theoretical analysis of
IPGG, we first conduct a theoretical analysis on IPD and provide
corresponding experimental validation. We model it based on the

|   | C | D |
|---|---|---|
| C | R, R | S, T |
| D | T, S | P, P |

Table 1: Matrix-form game

memory-1 type IPD game model (Foerster et al., 2018). The typical payoff matrix is as Table 1, where
$T > R > P > S$ and $2R > T + S$. Each participant has a discrete action space consisting of two
options: Cooperate (C) and Defect (D). The state space is defined as $s = [CC, CD, DC, DD, s_0]$,
where the first four elements represent the joint action outcomes from the previous round, and
$s_0$ denotes the initial state. Each episode lasts for 100 steps, with the payoff parameters set as
$[R, S, T, P] = [2, 0, 2.5, 1]$. We prove the following:

**Proposition 1.** *Two APC agents converge to mutual cooperation in the Iterated Prisoner's Dilemma.*

Let the actions of agents $i$ and $j$ be denoted by $a_i$ and $a_j$, where:

$$a_i \in \{C, D\}, \quad a_j \in \{C, D\}$$

The total reward for agent $i$ in APC is modified by punishment, where agent $i$ incurs a cost $w^{ij}c$
when punishing agent $j$ and agent $j$ incurs a penalty $w^{ij}\delta$ if they are punished for defection. So
the total reward for agent $i$ is: $r^i = R(a_i, a_j) - \sum_{j=1, j \neq i}^{N} w^{ij}c - \sum_{j=1, j \neq i}^{N} w^{ji}\delta$ Where $R(a_i, a_j)$
represents the payoff from the normal IPD payoff matrix, and the penalties are determined by the
defection detection and punishment.

The policy of each agent can be represented by 5 parameters, $\theta^a$, the probability of taking action
$C$ in each of these 5 states. In the case of IPD, these parameters correspond to the probability of
cooperation in the states $s_0, CC, CD, DC, DD$ as shown below:

$$\pi^a(C|s_0) = \theta^{a,0}, \quad \pi^a(D|s_0) = 1 - \theta^{a,0}$$

$$\pi^a(C|CC) = \theta^{a,1}, \quad \pi^a(D|CC) = 1 - \theta^{a,1}$$

$$\pi^a(C|CD) = \theta^{a,2}, \quad \pi^a(D|CD) = 1 - \theta^{a,2}$$

$$\pi^a(C|DC) = \theta^{a,3}, \quad \pi^a(D|DC) = 1 - \theta^{a,3}$$

$$\pi^a(C|DD) = \theta^{a,4}, \quad \pi^a(D|DD) = 1 - \theta^{a,4}$$

where $a \in \{1, 2\}$. We denote $\theta^a = (\theta^{a,0}, \theta^{a,1}, \theta^{a,2}, \theta^{a,3}, \theta^{a,4})$.

Let the distribution of $s_0$ as $p_0$ be:

$$p_0 = (\theta^1 \theta^2, \theta^1(1 - \theta^2), (1 - \theta^1)\theta^2, (1 - \theta^1)(1 - \theta^2))$$

Since punishment is executed in response to the opponent's defection, each action taken by the
agent—whether $C$ or $D$—may incur the same expected cost $c$ due to the possibility of the opponent
defecting. We denote this expected cost as $b$. Here, we assume that the PIW value assigned by the
DPN to a defection action is 1, which is consistent with our experimental results. So the payoff vector
influenced by punishment for each agent is given by:

$$r^1 = (R - b, S - b, T - \delta - b, P - \delta - b)^T, \quad r^2 = (R - b, T - \delta - b, S - b, P - \delta - b)^T$$

and the transition matrix is:

$$P = \left[ \theta^1 \theta^2, \theta^1(1 - \theta^2), (1 - \theta^1)\theta^2, (1 - \theta^1)(1 - \theta^2) \right]$$

Thus, the value functions for Agent 1 and Agent 2 can be written as:

$$V^1(\theta^1, \theta^2) = p_0^T \left( r^1 + \sum_{t=1}^{\infty} \gamma^t P^t r^1 \right)$$

$$V^2(\theta^1, \theta^2) = p_0^T \left( r^2 + \sum_{t=1}^{\infty} \gamma^t P^t r^2 \right)$$

Since $\gamma < 1$ and $P$ is a stochastic matrix, the infinite sum converges, and we have:

$$V^1(\theta^1, \theta^2) = p_0^T \left( \frac{I}{1 - \gamma P} r^1 \right), \quad V^2(\theta^1, \theta^2) = p_0^T \left( \frac{I}{1 - \gamma P} r^2 \right)$$

where $I$ is the identity matrix.

We describe the process by which Agent 1 updates its policy based on the reward structure and the incentives, and similarly for Agent 2.

Agent 1 updates its policy via the following update rule:

$$\hat{\theta}^1 = \theta^1 + \alpha \nabla_{\theta^1} V^1(\theta^1, \theta^2)$$

Expanding the value function $V^2(\theta^1, \theta^2)$:

$$\hat{\theta}^1 = \theta^1 + \frac{\alpha}{1 - \gamma} \nabla_{\theta^1} \Big( \theta^1 \theta^2 (R - b) + \theta^1 (1 - \theta^2)(S - b)$$
$$+ (1 - \theta^1)\theta^2 (T - \delta - b) + (1 - \theta^1)(1 - \theta^2)(P - \delta - b) \Big)$$
$$= \theta^1 + \frac{\alpha}{1 - \gamma} \Big( \theta^2 R + (1 - \theta^2)S - \theta^2 (T - \delta) - (1 - \theta^2)(P - \delta) - 2b \Big)$$

Thus, Agent 1's policy update becomes:

$$\hat{\theta}^1 = \theta^1 + \frac{\alpha}{1 - \gamma} \left( \theta^2 (R - S - T + P) + S - P + \delta \right)$$

And likewise for Agent 2:

$$\hat{\theta}^2 = \theta^2 + \frac{\alpha}{1 - \gamma} \left( \theta^1 (R - S - T + P) + S - P + \delta \right)$$

Substituting the values of $[R, S, T, P, \delta] = [2, 0, 2.5, 1, 2.6]$, we get:

$$(R - S - T + P) = (2 - 0 - 2.5 + 1) = 0.5 > 0$$

and

$$S - P + \delta = 0 - 1 + 2.6 = 1.6 > 0$$

Since $\theta^1 >= 0$ and $\theta^2 >= 0$, the entire process ensures that the cooperation rate continuously increases. This leads to the agents being able to escape from the inefficient Nash equilibrium and ultimately achieve cooperation.

Thus, the proposed APC method effectively promotes cooperation by making the short-term defection strategy less favorable, thereby converging towards a more optimal outcome for both agents.

We also conducted experiments on IPD, and the results are shown in Figure 9. The outcomes are consistent with those on IPGG: all agents escaped the dilemma of defection and achieved full cooperation, whereas the baselines failed to do so, resulting in full defection.

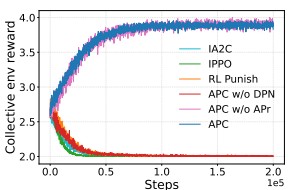

Figure 9: Learning curves for IPD of self-play training. The curves represent total collective rewards in environment.

**Differences between APC and traditional strategies.** APC differs from existing traditional strategies (like Tit-for-Tat(TFT), Cumulative Reciprocity (CR) (Li et al., 2022) or Win-Stay-Lose-Shift (WSLS)) in three key aspects. First, it is more effective at establishing cooperation and resisting exploitation. We propose TFT-D, which defects first and then mimics the opponent. For instance, in IPD, while TFT and TFT-D often fall into unproductive C–D cycles, APC can detect persistent defection and switch to defection itself, thereby both achieving cooperation with compatible partners and protecting against exploitation. Against policy (C, C, then alternating C/D), CR always C and gets exploited. WSLS also fails vs. Always Defect, alternating C/D after initial C. APC detects persistent defection and switches to D, protecting itself. Second, APC demonstrates stronger generalizability beyond the binary C/D decision space by handling complex real-world interactions. It adapts to cooperative partners while punishing defectors based on observed behaviors—for example, in SSH, where actions cannot be directly classified as cooperation or defection, APC can infer intent from its own observations and respond appropriately. Finally, APC exhibits greater proactiveness by actively

punishing defectors, thereby raising the cost of defection and incentivizing sustained cooperation in repeated interactions.

# B  ANALYSIS IN ITERATED PUBLIC GOODS GAME

## B.1  ANALYSIS OF COOPERATION PROMOTION

**Proposition 2.** Two APC agents converge to mutual cooperation in the Iterated Public Goods Game.

Compared to the IPD setting, IPGG involves a population of $n$ players. Instead of tracking the full joint action profile of other agents (which has $2^n$ possibilities), the state is defined by the number of contributors in the previous round. This reduces the state space from $2^n + 1$ to $n + 2$, greatly simplifying the state space.

Each agent $i \in \{1, 2, \ldots, n\}$ adopts a stochastic policy over actions:

$$\pi^i(a_i \mid s) = \begin{cases} \theta^{i,0}, & \text{if } a_i = c \text{ (contribute)} \\ \theta^{i,1}, & \text{if } a_i = d \text{ (not contribute)} \end{cases} \quad \text{subject to } \theta^{i,0} + \theta^{i,1} = 1, \ \theta^{i,0}, \theta^{i,1} > 0$$

Let the state $s = (p_c)$ be the proportion of contributors in the previous round. Then the transition to next state $s' = (p'_c)$ follows:

$$\mathbb{P}(s' \mid s; \theta) = \sum_{\substack{a_0, a_1, \ldots, a_{n-1} \\ \text{s.t. } n'_c}} \prod_{i=0}^{n-1} \pi^i(a_i \mid s) \quad \text{where } n'_c = \sum_{i=0}^{n-1} \mathbb{I}(a_i = c), \quad p'_c = \frac{n'_c}{n}$$

Define the population-level state distribution $p_0$ and the transition matrix $P \in \mathbb{R}^{(n+1) \times (n+1)}$ as a Markov chain over discrete cooperation levels $p_c \in \{0, \frac{1}{n}, \ldots, 1\}$. Let $r^i \in \mathbb{R}^{n+1}$ be the expected reward vector for agent $i$ under each $p_c$. Then the value function for agent $i$ is:

$$V^i(\theta) = p_0^T \left( \sum_{t=0}^{\infty} \gamma^t P^t r^i \right) = p_0^T \left( I - \gamma P \right)^{-1} r^i$$

Each agent $i$ updates its policy parameter $\theta^i$ by gradient ascent on the value function:

$$\theta^i \leftarrow \theta^i + \alpha \nabla_{\theta^i} V^i(\theta)$$

As in the IPD case, the gradient can be expanded similarly using the chain rule over $P$ and $r^i$, accounting for the influence of policy parameters on state transitions and expected payoffs. Let:

$$V^i(\theta) = V^i(\theta^i, \theta^{-i}) \quad \text{and} \quad \nabla_{\theta^i} V^i = \frac{\partial V^i}{\partial \theta^i} + \sum_{j \neq i} \frac{\partial V^i}{\partial \theta^j} \frac{\partial \theta^j}{\partial \theta^i}$$

Assuming independent updates, we approximate the gradient as:

$$\hat{\theta}^i = \theta^i + \frac{\alpha}{1 - \gamma} \cdot \nabla_{\theta^i} \left[ \theta^i \cdot r_c^i + (1 - \theta^i) \cdot r_d^i \right]$$

Here, we likewise assume that the PIW value assigned by the DPN to a defection action is 1, which is consistent with our experimental results. Since punishment in IPGG is executed in response to the opponent's defection, each action taken by the agent—whether contribute or not contribute-incur the same expected cost c due to the possibility of the opponent defecting. We also use $b$ to denote this expected cost. Substituting the expressions for rewards:

$$r_c^i = p_c \cdot \frac{(n-1)e \cdot r}{n} - e + \frac{e \cdot r}{n} - b, \quad r_d^i = p_c \cdot \frac{(n-1)e \cdot r}{n} - (n-1) \cdot \delta - b$$

So the policy update becomes:

$$\hat{\theta}^i = \theta^i + \frac{\alpha}{1 - \gamma} \left( \frac{e \cdot r}{n} - e + (n-1) \cdot \delta \right)$$

Since $[n, e, r, p] = [5, 1, 3, 1]$, this gradient is positive. Agent $i$'s cooperation probability $\theta^i$ increases with each update. Punishment ensures that defection becomes less attractive, promoting cooperation.

## B.2 Analysis of Penalty Parameter

Since the PIW value assigned by the DPN to a defection action is 1, it is reasonable to assume that all agents who choose to defect will be accurately punished by the APC agents. We choose the more complex IPGG environment to conduct a theoretical analysis of the punishment parameter. Under the assumption that all other environmental parameters remain fixed, we derive the following: The expected reward for an agent choosing to defect is given by $-4\delta$, while the reward for choosing to cooperate is $-0.4$.

Therefore, when $\delta > 0.1$, agents are incentivized to cooperate. Conversely, when $\delta < 0.1$, agents are more likely to defect. When $\delta$ is set to 0.1, cooperation and defection yield equivalent payoffs for the agents, making them indifferent between the two choices. Note in Figure 8 that when $\delta$ is set to 0.1, the cooperation rate is lower than 0.5. This may be because the punishment accuracy has not yet reached 100%, meaning that defectors are not always successfully penalized. As a result, in some cases, the expected return for defection may exceed that of cooperation, causing agents to gradually favor defection. Furthermore, when $\delta$ is set to 0.2 and $c$ is greater than 0.4, the cooperation rate remains low. This is likely because a high value of $c$ interrupts the learning process when $\delta$ is insufficient, weakening the incentive effect of punishment.

## B.3 Analysis of Penalty Parameters in Stochastic Sequential Games

This theoretical analysis of penalty parameter in IPGG can be extended to multi-step, non-tabular SSDs, using the Coingame as an example. We compare the relative magnitudes of the advantage functions $A(s, \pi_1)$ and $A(s, \pi_2)$ for an agent choosing between two policies at a given state: $\pi_1$ (picking a coin of another agent's color) and $\pi_2$ (picking a coin of its own color). Without APC intervention, $A(s, \pi_1) \approx A(s, \pi_2)$, so the probabilities of the agent choosing these two policies are similar. When APC is introduced, the action of picking a coin of another agent's color (denoted as $a^*$) incurs a penalty $\delta$, and the advantage function satisfies $A(s, \pi_1) \propto R_{\text{total}}(s, a^*) = 1 - \delta$. When $\delta > 1$, $A(s, \pi_1) < 0$, while $A(s, \pi_2) > A(s, \pi_1)$. As a result, the agent tends to prefer $\pi_2$ and avoids taking $\pi_1$. In our paper, $\delta = 1.1$ is set, satisfying $1 - \delta = -0.1 < 0$, which ensures $A(s, \pi_1) < 0$. Experimental results in Figure 4(a) show that, under this condition, the agent can effectively escape the dilemma in Coingame, consistent with the theoretical derivation.

## B.4 Analysis of Convergence and System Stability

**Proposition 3.** In the Iterated Public Goods Game, APC converges to a Nash equilibrium.

Consider $N$ agents, each with strategy $\pi_i$, and total expected payoff:

$$\mathbb{E}[\Pi_i^{\text{total}}] = \mathbb{E}[\Pi_i^{\text{PGG}}] - \sum_{i \neq j}(w_{ij}c + w_{ji}\delta)$$

Strategy update rule:

$$\pi_i^{(t+1)} = \pi_i^{(t)} + \alpha \cdot \nabla_{\pi_i}\mathbb{E}[\Pi_i^{\text{total}}]$$

Verify cross-derivative symmetry:

$$\frac{\partial \mathbb{E}[\Pi_i^{\text{total}}]}{\partial \pi_j} = -\delta\frac{\partial \mathbb{E}[w_{ji}]}{\partial \pi_j} - c\frac{\partial \mathbb{E}[w_{ij}]}{\partial \pi_j}$$

$$\frac{\partial \mathbb{E}[\Pi_j^{\text{total}}]}{\partial \pi_i} = -\delta\frac{\partial \mathbb{E}[w_{ij}]}{\partial \pi_i} - c\frac{\partial \mathbb{E}[w_{ji}]}{\partial \pi_i}$$

Since the penalty function $w_{ij} = f(\pi_i, \pi_j)$ satisfies:

$$\frac{\partial f(\pi_i, \pi_j)}{\partial \pi_j} = \frac{\partial f(\pi_j, \pi_i)}{\partial \pi_i}$$

the cross-derivatives are equal, making the system a potential game.

Construct potential function:

$$\Phi(\pi) = \sum_{i=1}^{N} \left( \mathbb{E}[\Pi_i^{\text{PGG}}] - \sum_{i \neq j} (w_{ij}c + w_{ji}\delta) \right)$$

which satisfies:

$$\frac{\partial \Phi}{\partial \pi_i} = \frac{\partial \mathbb{E}[\Pi_i^{\text{total}}]}{\partial \pi_i}$$

Rate of change of potential function:

$$\frac{d\Phi}{dt} = \sum_{i=1}^{N} \frac{\partial \Phi}{\partial \pi_i} \cdot \frac{d\pi_i}{dt} = \alpha \sum_{i=1}^{N} \left( \frac{\partial \Phi}{\partial \pi_i} \right)^2 \geq 0$$

Since $\Phi$ is bounded above and monotonically increasing, by Lyapunov stability theory, when $\frac{d\Phi}{dt} \to 0$, all $\frac{\partial \Phi}{\partial \pi_i} \to 0$, and the system converges to a Nash equilibrium.

We also present the theoretical analysis of system stability of APC:

Consider a homogeneous population of $N$ agents, all employing the APC algorithm. Each agent $i$ has an action space $a_i \in \{C, D\}$, and their strategy $\pi_i$ is updated via reinforcement learning to maximize expected return. The environment is an iterative public goods game with parameters $[n, e, r]$, satisfying $1 < r < n$.

The total payoff for agent $i$ at time $t$ is:

$$\Pi_i^{total}(t) = \Pi_i^{PGG}(t) - \sum_{j \neq i} w_{ji}(t)\delta - \sum_{j \neq i} w_{ij}(t)c$$

where $\Pi_i^{PGG}(t)$ is the public goods game payoff, $w_{ji}(t)$ is the received punishment intensity, $w_{ij}(t)$ is the imposed punishment intensity, $\delta$ is the unit punishment harm, and $c$ is the unit punishment cost.

Let $\rho_C(t)$ denote the proportion of cooperators, whose evolution satisfies:

$$\dot{\rho}_C(t) = \rho_D(t)W_{D \to C}(t) - \rho_C(t)W_{C \to D}(t)$$

The transition probabilities are determined by the expected payoff difference:

$$W_{D \to C} = \sigma(\mathbb{E}[\Pi_C] - \mathbb{E}[\Pi_D])$$

$$W_{C \to D} = \sigma(\mathbb{E}[\Pi_D] - \mathbb{E}[\Pi_C])$$

The expected payoffs for cooperators and defectors are respectively:

$$\Pi_C^{total} = \left( \frac{\rho(n-1)er}{n} - e + \frac{er}{n} \right) - (1 - \rho)(n-1)c$$

$$\Pi_D^{total} = \frac{\rho(n-1)er}{n} - (1 - \rho)(n-1)c - (n-1)\delta$$

The payoff difference is:

$$\Delta\Pi = \mathbb{E}[\Pi_C] - \mathbb{E}[\Pi_D] = -e + \frac{er}{n} + (n-1)\delta$$

At the steady state $\dot{\rho}_C = 0$, substituting the transition probabilities yields:

$$\rho_C^* = \sigma(\Delta\Pi)$$

When $\Delta\Pi = 0$, the system is at a critical state, solving for the critical punishment threshold:

$$\delta^* = \frac{e - \frac{er}{n}}{(n-1)}$$

When $\delta > \delta^*$, $\Delta\Pi > 0$ leads to $W_{D\to C} > W_{C\to D}$, and the system tends toward cooperation; when $\delta < \delta^*$, the system tends toward defection; when $\delta = \delta^*$, the system stabilizes at a mixed equilibrium.

### B.5    ANALYSIS OF SECOND-ORDER FREE-RIDER PROBLEM

**Proposition 4.** In the Iterated Public Goods Game, APC can avoid the second-order free-rider problem (i.e., cooperators' unwillingness to bear the cost of punishment leads to insufficient punishment provision) when the punishment intensity is sufficient.

The setting is set as follows: There are $n$ participants. Cooperators contribute $e$, and the amplification factor of the public good is $r$, satisfying $1 < r < n$. The strategy space includes pure cooperation $C$, defection $D$, and cooperation with punishment $P$. The overall proportion of cooperators is $\rho = x_C + x_P$. The punishment mechanism of APC is: P punishes D, P pays a cost $c$, and imposes a loss $\delta$ on D.

The payoff functions for each strategy are: Pure cooperator C: $\Pi_C = \frac{\rho(n-1)er}{n} - e + \frac{er}{n}$; Punisher P: $\Pi_P = \frac{\rho(n-1)er}{n} - e + \frac{er}{n} - (1-\rho)(n-1)c$; Defector D: $\Pi_D = \frac{\rho(n-1)er}{n} - x_P(n-1)\delta$

To suppress defection, consider making the payoffs of defectors D and pure cooperators C equal when the proportion of punishers is high. Set $\Pi_D = \Pi_C$ and substitute the payoff expressions:

$$\frac{\rho(n-1)er}{n} - x_P(n-1)\delta = \frac{\rho(n-1)er}{n} - e + \frac{er}{n}$$

Simplifying yields the critical punishment intensity:

$$\delta^* = \frac{e\left(1 - \frac{r}{n}\right)}{x_P(n-1)}$$

Based on the relationship between the actual punishment intensity $\delta$ and the critical value $\delta^*$, the system evolution can be divided into two scenarios, which we analyze using the replicator dynamics method:

The replicator dynamics equations describe how the growth rate of a strategy's frequency is proportional to its relative fitness (the difference between its payoff and the average payoff):

$$\dot{x_C} = x_C(\Pi_C - \bar{\Pi})$$
$$\dot{x_P} = x_P(\Pi_P - \bar{\Pi})$$
$$\dot{x_D} = x_D(\Pi_D - \bar{\Pi})$$

Since the three variables sum to 1, the system is effectively two-dimensional. We can use $x_P$ and $x_D$ as independent variables, so $x_C = 1 - x_P - x_D$.

Scenario 1: Dynamics under Sufficient Punishment ($\delta > \delta^*$). In this scenario, we assume the punishment intensity is strong enough to make defection unprofitable when the proportion of cooperators is high. Boundary equilibrium analysis: The all-cooperation edge $(x_P, x_D) = (x_P, 0)$, where $x_P \in [0, 1]$. When $x_D = 0$, the system lies on the edge formed by C and P. Here, $\rho = 1$, $\Pi_C = \Pi_P = e(r-1)$. According to the replicator dynamics equations, $\dot{x_C} = 0$ and $\dot{x_P} = 0$. This means the entire C-P edge (from all C to all P) consists of fixed points.

Stability analysis (using the Jacobian matrix): To determine the stability of these points on the C-P edge, we need to examine the system's dynamics after a small perturbation (i.e., introducing a small number of defectors D). Take any point $(x_P, x_D) = (x_P, 0)$ on the C-P edge, where $0 \le x_P \le 1$. Calculate the growth rate of the defector strategy at this point: $\dot{x_D} = x_D(\Pi_D - \bar{\Pi})$. When $x_D \to 0$, $\bar{\Pi} \approx \Pi_C = \Pi_P = e(r-1)$, and $\Pi_D = er - \frac{er}{n} - \rho(n-1)\delta$. Therefore, $\Pi_D - \bar{\Pi} = e - \frac{er}{n} - x_P(n-1)\delta$. For points on the C-P edge to be stable (i.e., resistant to invasion by D), we need $\Pi_D - \bar{\Pi} < 0$. This is equivalent to: $\delta > \frac{e\left(1 - \frac{r}{n}\right)}{x_P(n-1)} = \delta^*$. Thus, all points on the C-P edge are resistant to invasion by

defectors. At steady state, since defectors disappear, punishers no longer need to pay punishment costs, and their payoffs become identical to those of pure cooperators, thereby completely eliminating the second-order free-rider incentive.

Scenario 2: Dynamics under Insufficient Punishment ($\delta < \delta^*$). In this scenario, the punishment intensity is too weak to effectively suppress defection. Boundary equilibrium analysis: The full defection point $(x_P, x_D) = (0, 1)$ is a stable equilibrium (an attractor). For points on the all-cooperation edge $(x_P, x_D) = (x_P, 0)$, we calculate $\Pi_D - \bar{\Pi} = \frac{er}{n} - p(n-1)\delta$. Since $\delta$ is small, it is likely that $\Pi_D - \bar{\Pi} > 0$ for all $p \in [0, 1]$. Therefore, the entire C-P edge is unstable. Any tiny perturbation introducing defectors will drive the system away from the cooperative state because defectors obtain a higher payoff within a cooperative population.

Internal equilibrium analysis: The system might potentially have an internal equilibrium (where $x_C, x_P, x_D > 0$) where the payoffs of all three strategies are equal. This can be solved by simultaneously solving the equations $\Pi_C = \Pi_P = \Pi_D$. From $\Pi_C = \Pi_P$, we get: $\Pi_P - \Pi_C = -(1-\rho)(n-1)c = 0$. This implies that at equilibrium, we must have $\rho = 1$ or $c = 0$. Since $c > 0$, it must be that $\rho = 1$. But $\rho = 1$ implies $x_D = 0$, which contradicts the existence of an internal equilibrium ($x_D > 0$). Therefore, for $c > 0$, the system has no internal equilibrium. Pure cooperators C and punishers P cannot stably coexist long-term with defectors D while maintaining equal payoffs. Ultimately, the system converges to full defection, cooperation collapses, and the punishment mechanism fails.

## C  THE RELATIONSHIP BETWEEN THE DEFECTION PROBABILITY AND THE AGENT'S PAYOFF

**Proposition 5.** When the defection probability of an action is higher than the uniform probability, the payoff that this action brings to agent $i$ is lower than the average payoff caused by other actions to agent $i$. That is,

$$\sigma^{ij}(a^j) > \frac{1}{m} \quad \Rightarrow \quad U^i(a^j) < \bar{U}^i$$

Here, the action space size is $m = |\mathcal{A}^j|$, the average payoff is $\bar{U}^i = \frac{1}{m} \sum U^i(a^{j'})$, and $U^i(a^j) = r^i(o^i, a^j, a^{-j})$. In the following analysis, we will fix the observation $o^i$ and the actions of other agents $a^{-j}$.

Agent $i$ needs to estimate the defection probability $\sigma^{ij}(a^j)$ that agent $j$ takes action $a^j$, where $U^i(a^j)$ represents the payoff received by $i$. The training objective of defection prediction network $\mu^i$ is to maximize:

$$J(\mu^i) = \mathbb{E}_{a^j \sim \sigma^{ij}}[-U^i(a^j)] + \beta H(\sigma^{ij})$$

The first term encourages assigning high probability to low-payoff actions, and the second term is an entropy regularization term with weight $\beta > 0$, where $H(\sigma^{ij})$ is the Shannon entropy.

To solve the constrained optimization problem of maximizing $J(\mu^i)$ subject to the probability normalization constraint $\sum_{aj} \sigma^{ij}(a^j) = 1$, we formulate the Lagrangian as

$$\mathcal{L} = \sum_{aj} \sigma^{ij}(a^j)[-U^i(a^j)] - \beta \sum_{aj} \sigma^{ij}(a^j) \log \sigma^{ij}(a^j) + \lambda \left(1 - \sum_{aj} \sigma^{ij}(a^j)\right).$$

Solving this optimization involves taking the partial derivative with respect to $\sigma^{ij}(a^j)$, yielding

$$\frac{\partial \mathcal{L}}{\partial \sigma^{ij}(a^j)} = -U^i(a^j) - \beta \left(\log \sigma^{ij}(a^j) + 1\right) - \lambda.$$

Setting this derivative to zero gives the equation

$$-U^i(a^j) - \beta \left(\log \sigma^{ij}(a^j) + 1\right) - \lambda = 0.$$

Solving for $\sigma^{ij}(a^j)$, we find

$$\log \sigma^{ij}(a^j) = \frac{-U^i(a^j) - \lambda - \beta}{\beta},$$

which implies

$$\sigma^{ij}(a^j) \propto \exp\left(-\frac{U^i(a^j)}{\beta}\right).$$

Applying the normalization constraint, we obtain the final solution

$$\sigma^{ij}(a^j) = \frac{\exp(-U^i(a^j)/\beta)}{\sum_{a^{j'}} \exp(-U^i(a^{j'})/\beta)}$$

.

From the condition $\sigma^{ij}(a^j) > 1/m$, we obtain $\exp(-U^i(a^j)/\beta) > \bar{E}$, where $\bar{E} = \frac{1}{m}\sum_{a^{j'}} \exp(-U^i(a^{j'})/\beta)$.

Since $f(x) = \exp(-x/\beta)$ has a positive second derivative and is strictly convex, by Jensen's inequality:

$$\bar{E} = \frac{1}{m}\sum_{a^{j'}} f(U^i(a^{j'})) > f\left(\frac{1}{m}\sum_{a^{j'}} U^i(a^{j'})\right) = \exp(-\bar{U}^i/\beta)$$

$$\Rightarrow \exp(-U^i(a^j)/\beta) > \bar{E} > \exp(-\bar{U}^i/\beta)$$

Since $f(x) = \exp(-x/\beta)$ is a monotonically decreasing function, it follows that:

$$\Rightarrow U^i(a^j) < \bar{U}^i$$

## D   Analysis of Punishment Probability

**Proposition 6.** The punishment probability $p_t^{ij}$ converges to a stable equilibrium.

$p_t^{ij}$ is computed by Eq. (3). For more details, please refer to the description in the main text. Next, we will demonstrate its convergence. Modeling the problem as a stochastic process, define the state variable $X_t = p_t^{ij}$ as the punishment probability, $Y_t = f_t^{ij}$ as the defection frequency, and $\Delta Y_t = Y_t - Y_{t-1}$ as the frequency change. The update rule can be expressed as a Markov process: when $(\Delta Y_t \geq 0 \vee |Y_t - \bar{Y}_{t-1}| < \varepsilon) \wedge Y_t \geq \varepsilon$, $X_{t+1} = X_t - \frac{1}{m-1}$; otherwise, it remains unchanged.

Convergence analysis shows that $X_t \in [0, 1]$ is bounded, and the update rule is non-increasing. When the defection frequency $Y_t \to 0$, the system converges to a punishment probability $p^* \to 1$; when the defection frequency remains consistently high, $p^* \to 0$; and when periodic fluctuations occur, the system converges to a periodic steady state. According to the monotone convergence theorem, since $X_t$ is non-increasing and bounded below, it must converge to a random variable $X^*$. The limit value depends on the asymptotic behavior of $Y_t$: if $Y_t \to 0$, then $X^* \to 1$; if $Y_t$ remains high, then $X^* \to 0$; if $Y_t$ fluctuates periodically, it converges to a periodic solution, a property that can be proven by constructing a contraction mapping for the periodic mapping.

## E   Experiments

### E.1   Environment details

**Iterated Public Goods Game (IPGG).** In each round of the game, $n$ players independently decide whether to contribute their initial endowment $e$ in full to a common pool, without knowing the choices of others. The total contributions from all players are multiplied by a factor $r$ to form a "public good," which is then equally distributed among all players, regardless of whether they contributed. If a player chooses not to contribute, they retain their initial endowment. From a self-interested perspective, "not contributing" often becomes the more advantageous policy. Each episode lasts for 100 steps. The state includes the initial state as well as the historical contribution information of both the agent and other

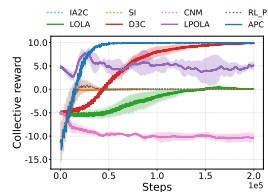

Figure 10:   Learning curves for IPGG of self-play training (including $\delta$ and $c$).

players. The action space consists of two discrete actions: Contribute (C) and Not Contribute (D). In our experiments, we adopt a commonly used parameter setting from public goods game studies: $[n, e, r] = [5, 1, 3]$.

**Coingame.** Two agents represent "Red" and "Blue" sides, respectively, and move freely on a 5×5 grid map to collect coins. At any given time, there is only one coin on the map, which is either red or blue. Once a coin is collected, a new coin is immediately generated at a random location with a randomly assigned color. Only one agent can occupy a grid cell at any time. The action space of each agent is $A = \{up, down, left, right, stay, pick up coin\}$. The environment state is represented as a tensor of shape $[5, 5, 5]$, consisting of five channels corresponding to Blue agent, Red agent, Blue coin, Red coin, and boundary mask. The first two channels encode the positions of the two agents, the last channel indicates whether a position is inside or outside the boundary, and the remaining two channels represent the current location of the coin (if present). Each episode lasts for 100 steps.

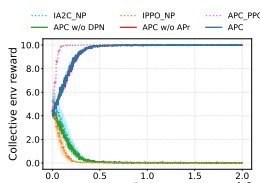

Figure 11: Learning curves for IPGG of self-play training.

**Sequential Snowdrift Game (SSG).** Four agents move freely on an 8×8 grid map. Snow piles can only be cleared when an agent moves to the location of a snow pile and performs the clear snow action. Once cleared, the snow does not regenerate. The action space of each agent is $A = \{up, down, left, right, stay, clear snow\}$. The state is represented as a tensor of shape $[6, 8, 8]$, consisting of six channels corresponding to the positions of Agents 1 through 4, the snow pile locations, and the boundary mask. Since agents who do not participate in clearing can still receive the maximum reward once the task is completed by others, self-interested agents tend to free-ride rather than actively clearing snow. Each episode lasts for 50 steps.

**Sequential Stag-Hunt (SSH).** SSH is defined as an 8×8 two-dimensional grid, within which four agents can move freely across the cells. Whenever an agent successfully completes a hunt, it immediately exits the environment; captured prey does not respawn. To catch a hare, an agent must move to the hare's location and perform the hare-hunting action. To hunt a stag, two agents must simultaneously reach the stag's location and both perform the stag-hunting action. Each agent's action space of is $A = \{up, down, left, right, stay, hunt hare, hunt stag\}$. If an agent hunts a hare alone, the other agents who do not hunt hares incur a penalty of 0.1 points due to the defection of hare-hunting agent. The state is a tensor of shape $[7, 8, 8]$, with seven channels for four agents, hare, stag, and boundary mask. Each episode lasts for 30 steps.

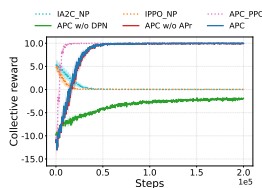

Figure 12: Learning curves for IPGG of self-play training (including $\delta$ and $c$).

**Foraging.** The 12 agents in the Foraging game move freely across its 10x10 grid map. Common agents that can only collect common berries, while special agents are capable of collecting both common berries and forbidden berries. If multiple agents move to the same cell for harvesting, the reward is divided equally among them. The environment initially comprises 50 berries, 70% of which are common and have a normal yield of 3 points. However, once a special agent collects a forbidden berry, it triggers permanent resource degradation, reducing the reward of all common berries to 1 point, and negatively affecting the rewards of all common agents with 0.1 point. Each time a berry is collected, a new berry regenerates, with a 0.1 probability of being a forbidden berry and the rest being common berries. To enforce norms, the system includes a marking and punishment mechanism: agents that violate the norm by collecting forbidden berries are marked for 10 time steps. Common agents can spend a cost of $c$ points to punish other agent, forcing the latter to pay a fine of $\delta$ points. Each agent's action space of is $A = \{up, down, left, right, stay, pick berry\}$. The state is a tensor of shape $[16, 10, 10]$, with sixteen channels for 12 agents, common berry, forbidden berry, marked signal and boundary mask. Each episode lasts for 50 steps.

In the specific experiments, the proposed method introduces additional punishment parameters, and corresponding punishment parameters $[\delta, c]$ are set in each environment. The parameter configurations for each environment are as follows: in IPD, they are set to [2.6, 2.6]; in IPGG, they are [0.7, 0.7]; in Coingame, they are [1.1, 1.1]; in SSG, they are [2.1, 2.1]; in SSH, they are [0.4, 0.4]; and in Foraging, they are [0.7, 0.7]. These parameters are used to measure the fines imposed on others by penalties and the costs incurred by the agent. To discourage defection, $\delta$ is chosen such that the expected

| Parameter | SSDs Value | IPD/IPGG Value | Parameter | SSDs Value | IPD/IPGG Value |
|---|---|---|---|---|---|
| $\alpha_\theta$ | 5e-4 | 1e-3 | $\alpha_\mu$ | 5e-5 | 1e-5 |
| $\gamma$ | 0.99 | 0.99 | parr_num | 50 | 10 |
| buffer_size | 3e4 | 3e4 | batch_size | 1e4 | 1e3 |
| $L$ | 2e5 | 2e3 | $\varepsilon$ | 1e-4 | 1e-2 |
| $\beta$ | 8e-2 | 1e-3 | | | |

Table 2: Hyperparameters for SSDs and IPD. These values represent the settings used in the experiments for SSDs and IPD, comparing values in both cases. Buffer_size refers to the total amount of data collected for training the defection action network, and batch_size denotes the amount of data used when training the policy network.

punishment offsets the potential gain from defection. $c$ incurred by the punisher is set equal to $\delta$ applied. The details of the hyperparameters are provided in Table 2.

### E.2 IMPLEMENTATIONS

**SSDs.** The actor network is responsible for selecting actions based on the current state of the environment. It starts with two convolutional layers, each using 3x3 kernels with strides of 1 and padding of 1. These layers process input multi-channel data, with the output being passed through a fully connected (FC) layer, which has 128 units and is activated using the ReLU function. The processed output is then passed to an LSTM cell, which maintains the internal state of the agent. Finally, the LSTM output is passed to the actor head, which consists of a linear layer that produces the action logits. A softmax activation is applied to these logits to generate the probability distribution over actions.The network uses Xavier uniform initialization for the weights and zero initialization for the biases in the linear layers. The action logits corresponding to the actionmask are set to negative infinity to prevent invalid actions from being selected.

The critic network estimates the value function for a given state. Like the physical actor network, it uses two convolutional layers to process the input data. After the convolutional layers, the output is passed through a fully connected layer with 128 units and a ReLU activation. The output of this layer is then passed to another fully connected layer, which outputs the estimated state value. The network uses Xavier uniform initialization for the weights and zero initialization for the biases. Unlike the actor network, the critic network does not include an LSTM layer, as it focuses on estimating the value rather than selecting actions.

The defection predictor network is designed to estimate the rewards based on the current state and the actions taken by the agent. The dimensionality of the actions taken by the agent is fixed, and the actions of agents outside the observation range are represented using a default placeholder value of -1. Similar to the actor and critic networks, the network starts with two convolutional layers for multi-channel data processing. The output of these layers is then passed through a fully connected layer with 128 units and ReLU activation. The network also takes the actions taken by other agents as input, concatenating this with the processed state representation. The combined input is passed to the fully connected layer, followed by an output layer that produces action logits. The network uses the softmax activation to convert the action logits into a probability distribution over the possible actions. The final output of the network is the action probabilities, which represent the agent's decision about the next action based on the current state and the actions of other agents. It also uses Xavier uniform initialization for weights and zero initialization for biases in its linear layers.

**IPD and IPGG.** All the CNNs and LSTM in IPD and IPGG are removed. The size of FC layer is scaled down to 32. In IPD, the state consists of the action choices made by both agents in the previous round. In contrast, in IPGG, the state is defined by the number of contributors and non-contributors in the previous round. As a result, the state dimensions in both IPD and IPGG remain consistent.

**Experiments Compute Resources**

The experiments were conducted on a system equipped with 24 13th Gen Intel(R) Core(TM) i9-13900KF CPUs running at 800MHz. The machine is equipped with one NVIDIA GeForce RTX 4090 GPU, with 63,488 MiB of memory. The operating system used is Ubuntu 22.04.

## F  ADDITIONAL RESULTS

### F.1  SUPPLEMENT TO MAIN RESULTS

**Collective Rewards during Self-Play Training in IPGG and four SSDs of main results.** We have supplemented the results showing the collective reward including punishment impact, as shown in Figure 10 and 13. This considers not only the original environment reward but also the effects of $\delta$ and $c$. Both LPOLA and CNM sometimes become trapped in the influence of pseudo-rewards, which prevents them from effectively optimizing the true environmental rewards. Other methods quickly learn to avoid punishment once it perceives associated cost, thus failing to make effective use of the punishment. Notably, the APC approach maintains collective reward levels comparable to those in the original environment even after considering $\delta$ and $c$, demonstrating that it has successfully learned to use punishment policy appropriately: punishing in response to defection and avoiding unnecessary punishment in cooperative scenarios, thereby maximizing the effectiveness of punishment.

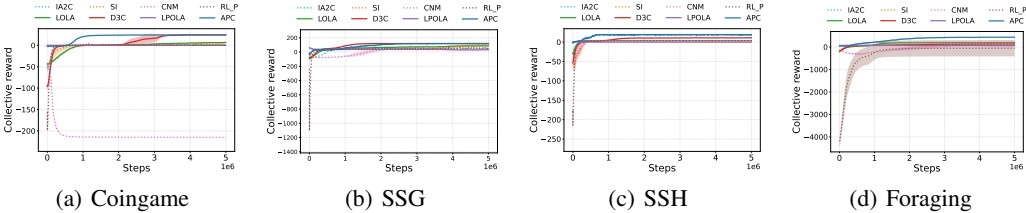

|   (a) Coingame   |   (b) SSG   |   (c) SSH   |   (d) Foraging   |

Figure 13: Collective rewards for four SSDs of self-play training (including $\delta$ and $c$).

**Collective Rewards under Different Initial Strategies in SSH.** To better evaluate reality-resembling and context-dependent scenarios, we add 14 additional experimental configurations in the 4-agent SSH environment, where each agent is initialized as one of three behavioral types: a stag_hunting agent (S), which always hunts stag; a hare_hunting agent (H), which always hunts hare; or a random agent (R), which selects actions randomly. These configurations cover a wide variety of initial conditions: we vary the ratios of stag_hunting and random agents (1S3R, 2S2R, 3S1R, 4S) and hare_hunting and random agents (1H3R, 2H2R, 3H1R, 4H); consider combinations of stag_hunting and hare_hunting agents only (1S3H, 2S2H, 3S1H); and include mixed-type configurations involving all three agent types (1S1H2R, 1S2H1R, 2S1H1R). All 14 configurations are trained using our proposed APC method. To evaluate how well each method approaches the theoretical optimum, we use the ratio between the achieved total reward and the maximum possible reward. A ratio closer to 1 indicates better performance. The results are summarized in Table 3.

|       | 1S3R  | 2S2R  | 3S1R  | 4S    | 1H3R   | 2H2R   | 3H1R   |
|-------|-------|-------|-------|-------|--------|--------|--------|
| Ratio | 0.949 | 0.967 | 0.961 | 0.968 | 0.957  | 0.955  | 0.947  |
|       | 4H    | 1S3H  | 2S2H  | 3S1H  | 1S1H2R | 1S2H1R | 2S1H1R |
| Ratio | 0.934 | 0.970 | 0.936 | 0.961 | 0.952  | 0.947  | 0.946  |

Table 3: Performance ratio of the APC method under different 4-agent SSH configurations. Each value represents the ratio between the achieved collective rewards and the theoretical maximum.

Experimental results demonstrate that, under the APC framework, cooperation consistently emerges via the punishment mechanism, regardless of agents' initial strategies or types. In particular, even in adversarial contexts such as the 3H1R setting, agents initially favoring individualistic strategies (e.g., hunting hares) can modify their policies and ultimately choose to cooperate in hunting the stag. This indicates that our method can effectively bootstrap cooperative behavior from non-cooperative initial conditions, highlighting its robustness in promoting cooperation across diverse agent configurations.

**Personal Rewards during Self-Play Training in IPGG and SSH.** We selected the results from the remaining four random seeds for presentation. For clarity, we show only the results from two random seeds. The results are presented in the Figure 14 and 15. For these two random seeds, the outcomes are largely consistent with those observed for the random seed in Figure 5. Notably, in CRM, the agents' division of labor differs, whereas APC continues to exhibit exceptionally high fairness.

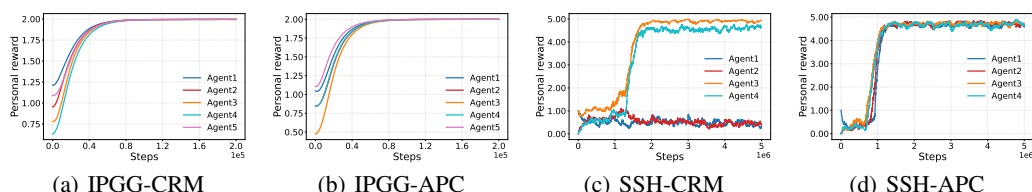

(a) IPGG-CRM      (b) IPGG-APC      (c) SSH-CRM      (d) SSH-APC

Figure 14: Learning curves of each agent in IPGG and SSH for CRM and APC, second random seed.

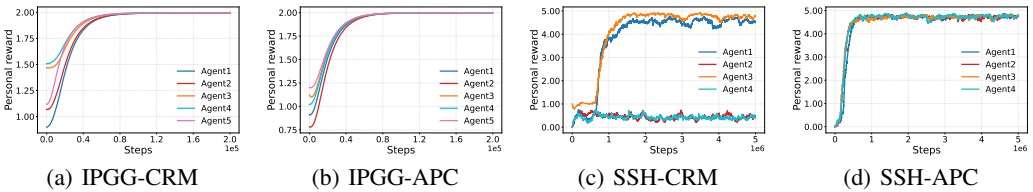

(a) IPGG-CRM      (b) IPGG-APC      (c) SSH-CRM      (d) SSH-APC

Figure 15: Learning curves of each agent in IPGG and SSH for CRM and APC, third random seed.

## F.2 SUPPLEMENT TO ABLATION STUDY

**Collective Environmental Rewards and Collective Rewards during Self-Play Training in IPGG and four SSDs of Ablation Experiment.** We supplement the performance of IA2C and IPPO in SSDs without the punish action (labeled IA2C_NP and IPPO_NP), as shown in the Figure 11, 12, 16 and 17. Here, IPPO refers to Independent Proximal Policy Optimization, which is another commonly used independent reinforcement learning method that introduces a clipping mechanism during policy updates. This approach enables more stable policy optimization and improves both training stability and sample efficiency (Schulman et al., 2017). The results indicate that simply adding this punitive action provides no performance improvement, as IA2C performs just as poorly as the original IA2C_NP. We also present the performance of APC w/o DPN and APC w/o APr in terms of collective reward, as shown in the Figure 11, 12, 16 and 17. The experimental results indicate that although APC w/o DPN can avoid excessive punishment through adaptive punishment, it fails to implement targeted punishment against defectors and therefore does not improve overall performance. Meanwhile, APC w/o APr achieves collective rewards comparable to APC, but fails to effectively respond to rule-based agents, as demonstrated in Figure 6(a) and 6(c). To further verify the generality of APC, we substituted the A2C algorithm in APC with PPO. The resulting APC_PPO framework yielded a comparable performance gain, as shown in the Figure 11, 12, 16 and 17, showing that our method's effectiveness is robust to the underlying RL algorithm.

## F.3 SUPPLEMENT TO ADAPTIVE PUNISHMENT ABILITY

**Collective Rewards vs Self-Interested PPO Agents in IPGG and SSH.** To investigate the ability of APC to shape other self-interested yet learning agents, we let a focal APC agent interact with PPO agents in the IPGG and SSH environments. To ensure fairness, the PPO agents were also endowed with the ability to punish others, equipped with APC 's adaptive punishment module. Otherwise, since the APC agent is not constrained by punishment, it would naturally choose to exploit the PPO agents by defecting. The experimental results, shown in Figure 18(a) and 18(b), indicate that the collective rewards in both the IPGG and SSH environments are consistent with those presented in Figure 10 and 13(c). In IPGG, agents achieve full cooperation, with all individuals choosing to contribute, while in SSH, agents learn to cooperate in hunting stag. These results show that APC can leverage punishment to guide self-interested agents from defection towards cooperation.

**Collective Rewards during Self-Play Training in MIPGG and MSSH.** We evaluated the collective rewards of APC (including $\delta$ and $c$) in the MIPGG and MSSH environments in Figure 18(c) and 18(d). The final results are largely consistent with those shown in Figure 10 and 13(c). Although the larger action space slightly slows down the learning process, the final converged reward values remain consistent. This indicates that our method remains effective in environments with multiple defection actions, promoting cooperation and enancing collective rewards. Specially, in MIPGG and MSSH, the punishment parameter $\delta$ is set to 2.1 and 3, respectively. As the number of defection actions

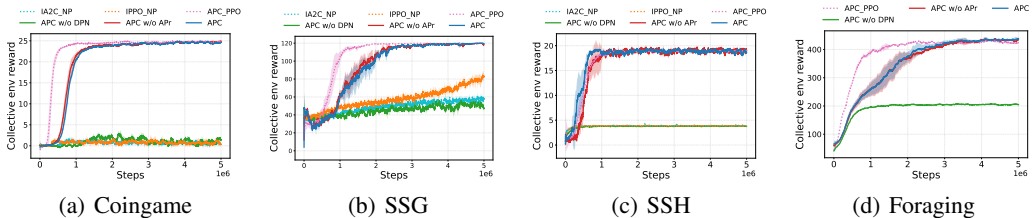

(a) Coingame     (b) SSG     (c) SSH     (d) Foraging

Figure 16: Collective environmental rewards for four SSDs of self-play training of ablation experiments.

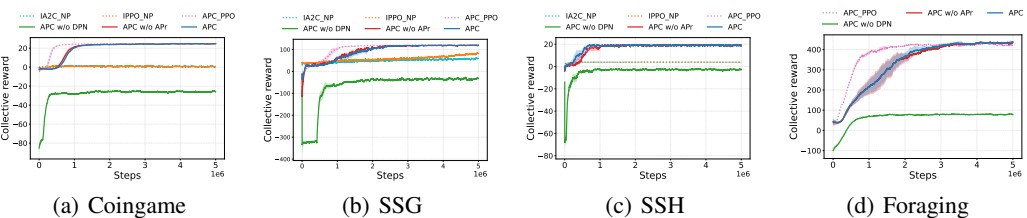

(a) Coingame     (b) SSG     (c) SSH     (d) Foraging

Figure 17: Collective rewards for four SSDs of self-play training of ablation experiments (including $\delta$ and $c$).

increases, a higher punishment intensity is required to maintain cooperation, because actions with lower levels of defection may not be sufficiently deterred if the punishment is too weak.

## F.4 SUPPLEMENT TO HYPERPARAMETER SENSITIVITY

**Collective Rewards during Self-Play Training with different $\beta$ in MSSH.** In Figure 8(b), we evaluate the effect of different $\beta$ values on DPN recognition results in MSSH. The results show that while $\beta$ slightly changes the punishment weights for different levels of defection, the overall trend remains consistent. To further examine the impact of these subtle differences on the collective reward, we conducted self-play training of APC in MSSH with $\beta$ set to 0.001, 0.005, and 0.01. The experiment results in Figure 19 indicate that a smaller $\beta$ corresponds to weaker punishment intensity, leading to lower punishment costs during the initial stage of regulating opponents' behaviors.

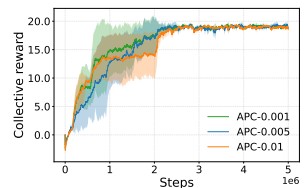

Figure 19: Learning curves for MSSH of self-play training with different $\beta$.

However, as opponents gradually adjust and adopt cooperative strategies, the system converges to cooperation, and the final collective reward approaches the theoretical optimum. This demonstrates that although different $\beta$ values cause minor variations in punishment intensity, they have little effect on the ultimate collective reward.

## F.5 ACCURACY AND GENERALIZATION OF DEFECTION PREDICTION NETWORK

To evaluate the identification accuracy of the DPN, we monitored its prediction accuracy in SSH. The configuration of the DPN during training is as follows. During each DPN training session, the dataset remains fixed, resulting in a stable data distribution, which ensures its convergence. When newly collected data is introduced, to maintain consistency in data distribution, we extract a portion of the data previously gathered by a random policy from the buffer and mix it with the newly collected data in a 1:1 ratio for DPN training. This approach enables the assimilation of new data while preserving policy diversity—guaranteed by the data collected through the random policy—thereby exposing the network to a wider range of potential policy combinations. This ensures model accuracy while enhancing its adaptability to novel opponent strategies. Experimental results show that even when using a DPN model trained in an earlier phase to evaluate opponent behavior under new policies, its prediction performance remains stable, with an average accuracy of 97.674% and a variance of 1.792. This indicates that the model learned by the DPN possesses strong generalization capability and robustness.

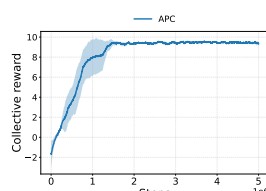

Figure 20: Learning curves for SSH with different reward structures of self-play training (including $\delta$ and $c$).

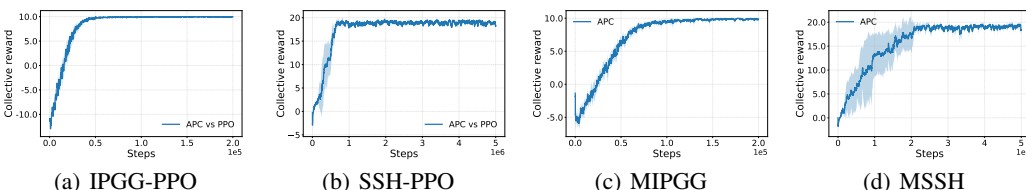

|  (a) IPGG-PPO  |  (b) SSH-PPO  |  (c) MIPGG  |  (d) MSSH  |

Figure 18: Collective rewards of a APC agent vs PPO in IPGG and SSH, and of self-play training in MIPGG and MSSH. (a) and (b) show the performance when a focal APC agent interacts with self-interested PPO agents in IPGG and SSH. (c) and (d) present collective rewards during self-play training in MIPGG and MSSH, which feature a larger action space and more defection options.

To evaluate the identification accuracy of the converged DPN, we conducted experiment in SSH. In SSH, defection is defined as the act of hunting hare when the other agent is hunting stag. We collected 50,000 trajectory data points using a random policy. These trajectories were fed into the pre-trained defection prediction network, which outputs a probability distribution over the target agent's actions based on the current observation and the other agent's action. An action is classified as defection when the probability assigned by the network exceeds that of a uniform distribution over the action space. By comparing the network's predictions with the ground-truth defection labels, we calculated the identification accuracy. Experimental results show that the defection prediction network achieved an accuracy of 98.975% in the SSH environment. This demonstrates that the defection prediction network can effectively identify defection behaviors across different environments, providing a reliable foundation for the subsequent punishment mechanism.

**Collective Rewards in SSH with different reward structures.** In SSH, when the reward for hunting stag is adjusted to 5, the relative influence of the opponent's actions on our rewards remains consistent. Under such conditions, the trained DPN can be directly transferred and still achieve full cooperation in stag hunting show in Figure 20. This demonstrates that the trained DPN retains generalization capability when encountering opponents or environments with differing reward structures.

## G  USE OF LLMS

In this work, we utilized a large language model (LLM), specifically GPT-4o, to support certain aspects of the research process. The model was employed for text editing, including grammar correction, spelling checks, and improvement of word choice and sentence structure, which helped enhance the clarity and readability of the manuscript. In addition, the LLM was used to assist in visualizing results, facilitating the creation of clear and well-structured figures and tables for inclusion in the paper. It is important to note that all scientific ideas, experimental designs, and data analyses were conducted by ourselves, and the LLM was only used as a tool to improve presentation quality and visualization.

## H  BROADER IMPACT

The APC framework proposed in this study holds significant implications for the emergence of social behavioral norms in multi-agent systems. By introducing an adaptive punishment method that does not rely on additional reward resources, APC offers a feasible path to reconcile individual rationality with collective optimality, helping to address the prevalent defection–cooperation dilemma in artificial intelligence systems. In real-world scenarios such as complex social decision-making systems, autonomous traffic coordination, collaborative robotics, and virtual economies, this method has the potential to enhance overall system efficiency, stability, and fairness.

Moreover, APC embodies a decentralized and self-organizing method for cooperation formation through autonomous defection recognition and rational punishment policy. This provides technical insights for research on social contracts, norm evolution, and AI ethics, contributing to the development of intelligent systems with value alignment and social responsibility. However, if the punishment is misused or if the training data is biased, it may lead to unjust punishment or incentive imbalance. Therefore, when deploying such systems, attention should be paid to the interpretability,

accountability, and fairness of the punishment to ensure that it serves the public good without causing unreasonable harm to individuals.

---

**Algorithm 1** Adaptive Punishment for Cooperation (APC)

---

1: **Input:** Environment $M$, number of agents $N$, hyperparameters $\beta, c, \delta, L, \varepsilon$
2: Initialize policy networks $\pi^i_{\theta^i}$, replay buffers $\mathcal{D}^i$ for $i = 1, \ldots, N$ collected by random policy, defection prediction networks $\mu^i$ trained by $\mathcal{D}^i$
3: Initialize punishment probabilities $p^{ij} \leftarrow 1$ for all $i, j$
4: Initialize defection frequency history $f^{ij} \leftarrow \emptyset$
5: **Training Phase:**
6: **for** episode $= 1, 2, \ldots,$ max_episodes **do**
7:      Collect trajectories by $\pi_\theta$ and Store $(o^i_t, \vec{a}_t, r^i_t)$ in $\mathcal{D}^i$
8:      **for** each agent $i$ **do**
9:          For each transition in batch:
10:              Compute defection distribution $\sigma^{ij}_t = \mu^i(o^i_t, a^{-j}_t)$
11:              Compute punishment weight $w^{ij}_t$ using Eq.(4) and $p^{ij}$
12:              Compute total reward:

$$r^{i,\text{tot}}_t = r^i_t - \sum_{j \neq i} w^{ij}_t c - \sum_{j \neq i} w^{ji}_t \delta$$

13:          Update $\pi^i_{\theta^i}$ using A2C with $r^{i,\text{tot}}_t$
14:      **end for**
15:      **for** each pair $(i, j)$ **do**
16:          Compute defection frequency $f^{ij}_s$ over last $L$ steps
17:          Update $p^{ij}$ using Eq.(3):
18:      **end for**
19:      **if** episode mod $update\_frequency = 0$ **then**
20:          **for** each agent $i$ **do**
21:              **train** $\mu^i$ to maximize Eq.(2)
22:          **end for**
23:      **end if**
24: **end for**
25: **Execution Phase:**
26: **for** time step $t = 1, 2, \ldots, T$ **do**
27:      **for** each agent $i = 1, \ldots, N$ **do**
28:          Observe current observation $o^i_t$
29:          Select action $a^i_t$ according to policy $\pi^i_{\theta^i}$
30:          **for** each other agent $j \neq i$ **do**
31:              Compute punishment intensity $w^{ij}_t$ (according to Eq.(4) and $p^{ij}$)
32:              Update $p^{ij}$ using Eq.(3)
33:          **end for**
34:      **end for**
35:      Execute joint action $\vec{a}_t = (a^1_t, \ldots, a^N_t)$ and $\vec{w}_t = (w^1_t, \ldots, w^N_t)$
36:      Environment returns individual rewards $r^{i,\text{tot}}_t$ and new observations $o^i_{t+1}$
37:      Compute defection frequency $f^{ij}_s$ over last $L$ steps
38: **end for**

---

