# OpenReview forum: "Adaptive Punishment for Cooperation in Mixed-Motive Games"
_ICLR.cc/2026/Conference — Submitted to ICLR 2026_

### Official Review · Reviewer_duR2 · 2025-10-23

**Soundness:** 2
**Presentation:** 2
**Contribution:** 2
**Rating:** 4
**Confidence:** 4

**Summary:**

This paper introduces a distributed multi-agent reinforcement learning framework called Adaptive Punishment for Cooperation (APC), which aims to promote cooperation among self-interested agents in mixed-motive environments. The approach integrates two components: a defection awareness module that learns to recognize and quantify the severity of opponents’ defections based on observed behavior and rewards, and an adaptive punishment mechanism that dynamically adjusts both the probability and intensity of sanctions according to their past effectiveness. This enables agents to punish selectively—strongly when defections are harmful, and lightly or not at all when punishment fails to reduce defection—balancing the cost of enforcement with its benefits. Theoretical analysis and experiments on multiple benchmark environments, including Iterated Public Goods Game and several sequential social dilemmas, demonstrate that APC achieves high levels of cooperation, fairness, and robustness, outperforming standard reinforcement learning baselines and fixed-rule punishment schemes. The work provides new insight into how adaptive, context-aware sanctioning can stabilize cooperation in decentralized multi-agent systems.

**Strengths:**

1. This paper proposes the first decentralized adaptive punishment mechanism that simultaneously adjusts both probability and intensity of punishment according to defection severity and historical effectiveness. Also, it introduces a Defection Predictor Network (DPN) — a learned defection-awareness model that quantifies opponent defection rather than using rule-based detection, marking a significant step beyond static or heuristic punishment systems in MARL. Therefore, it bridges social norm enforcement and multi-agent reinforcement learning through an explicitly learnable and theoretically grounded mechanism.
2. This paper provides a mathematical formulation of adaptive punishment, including explicit update rules for punishment probability $p_{ij,t}$​ and intensity $w_{ij,t}$​. The theoretical analysis clarifies conditions under which cooperation becomes a rational equilibrium (e.g., when penalty $\delta > 0.1$), linking empirical results to formal reasoning. Carefully distinguishing between defection severity detection (via DPN) and punishment decision-making (via adaptive probability), shows strong internal logic and interpretability.
3. The proposed approach is evaluated across four distinct social dilemma environments — Iterated Public Goods Game (IPGG), Coingame, Sequential Snowdrift Game (SSG), and Sequential Stag-Hunt (SSH)—covering both classical and spatiotemporal settings. The experimental result demonstrates substantial improvements in cooperation rate, fairness, and robustness compared to IA2C, IPPO, and RL-Punish baselines.
4. The proposed cost-efficient mechanism could have some realistic implication: reduces unnecessary punishment and stabilizes cooperation without requiring a central authority or added reward resources.
5. This paper is generally well-structured exposition follows the logic: conceptual motivation → formalism → algorithm → theoretical and empirical validation. Visualizations (Figures 3–7) effectively illustrate performance, fairness, and sensitivity trends.

**Weaknesses:**

1. All four tested environments (IPGG, Coingame, Sequential Snowdrift, Sequential Stag-Hunt) are toy or small-scale social dilemmas with simple discrete actions.
2. No analysis is provided on whether DPNs converge to stable opponent models or oscillate due to non-stationarity. This raises concerns about reproducibility in other scenarios.
3. The theoretical justification (e.g., δ > 0.1 ⇒ cooperation rational) is derived under idealized assumptions that don’t hold in stochastic sequential games. The analysis is more illustrative than rigorous, limiting theoretical strength.
4. Baselines (IA2C, IPPO, RL-Punish) are relatively weak. Other relevant baselines required to highlight the performance and significance of the proposed approach, e.g., LOLA [1], D3C [2] and Social Influence [3].

[1] Foerster, J., Chen, R. Y., Al-Shedivat, M., Whiteson, S., Abbeel, P., & Mordatch, I. (2018). _Learning with Opponent-Learning Awareness (LOLA)._ In Proceedings of the 17th International Conference on Autonomous Agents and Multiagent Systems (AAMAS).

[2] Gemp, I., McKee, K. R., Everett, R., Duéñez-Guzmán, E., Bachrach, Y., Balduzzi, D., & Tacchetti, A. (2022). _D3C: Reducing the Price of Anarchy in Multi-Agent Learning._ In Proceedings of AAMAS 2022.

[3] Jaques, N., Lazaridou, A., Hughes, E., Gulcehre, Ç., Ortega, P. A., Strouse, D. J., Leibo, J. Z., & de Freitas, N. (2019). _Social Influence as Intrinsic Motivation for Multi-Agent Deep Reinforcement Learning._ In Proceedings of the 36th International Conference on Machine Learning (ICML).

**Questions:**

Please address the weaknesses and answer the following technical questions:
1. The punishment probability $p_{ij,t}$ is adjusted using defection frequency over rolling windows via Eq. (3). Is there any theoretical guarantee that this stochastic update rule converges to a stable equilibrium (e.g., a fixed point or bounded oscillation)? How sensitive is it to window length L and threshold ε ?
2. The DPN learns to predict defection severity via a proxy objective involving $−r_i(t)$ and entropy regularization. Since this optimization does not use ground-truth defection labels, how do the authors ensure that the predictor does not conflate noise or random fluctuations in rewards with genuine defection behaviors? Can the learned DPN generalize across different opponents or environments with altered reward structures?
3. Fairness is evaluated using the equality metric $E = 1 − (\sum_i \sum_j |R_i − R_j|) / (2N \sum_i R_i)$. Is this metric appropriate when negative rewards or punishments dominate? How does it behave when collective reward is close to zero (denominator instability)?
4. The paper provides an analytical result that under APC, expected defection reward is −4δ and cooperation reward is −0.4. How were these values derived, and do they hold generally across all parameter ranges or only under specific assumptions of IPGG? Is this analysis extendable to multi-step, non-tabular SSDs?

---

> ### Author Response · Authors · 2025-11-25
> **Response to Reviewer duR2 (1)**
>
> Thank you very much for the thoughtful and constructive comments. We’re grateful for the opportunity to clarify and improve our work, and we’ve carefully addressed each point below.
> >W1: All four tested environments (IPGG, Coingame, Sequential Snowdrift, Sequential Stag-Hunt) are toy or small-scale social dilemmas with simple discrete actions.
>
> We introduced a more complex, larger-scale heterogeneous Foraging environment involving 12 agents. The environment setup is based on the paper referenced as [1]. In Foraging, 12 agents move across a 2D grid and collect resources. Among them, 75% are common agents that can only collect common berries, while 25\% are special agents that can also collect forbidden berries (4 points). Initially, 70% of the resources in the environment are common berries, normally yielding a reward of 3 points. However, once a special agent collects a forbidden berry, it triggers permanent resource degradation, reducing the reward of all common berries to 1 point, and negatively affecting the rewards of all common agents. Each time a berry is collected, a new berry regenerates, with a 0.1 probability of being a forbidden berry and the rest being common berries. To enforce norms, the system includes a marking and punishment mechanism: agents that violate the norm by collecting forbidden berries are marked for 10 time steps. common agents can spend a cost of 0.7 points to punish other agent, forcing the latter to pay a fine of 0.7 points.
>
> We compared the APC method with other baselines, and the experimental results show that in this larger-scale heterogeneous environment, APC still achieves superior performance compared to the baselines. It effectively utilizes punishment to deter special agents from collecting forbidden berries, encouraging them to collect common berries instead, thereby enhancing collective returns.
>
> |Collective env reward/Collective reward|Foraging|
> |:-:|:-:|
> |**LOLA**|$191.630(\pm5.446)/ 191.621(\pm5.436)$|
> |**SI**|$193.783(\pm6.223)/ 193.234(\pm5.981)$|
> |**D3C**|$162.443(\pm10.634)/ 162.035(\pm10.146)$|
> |**LPOLA**|$60.645(\pm5.406)/ 60.645(\pm5.406)$|
> |**CNM**|$207.828(\pm0.846)/128.887(\pm12.176)$|
> |**APC**|$439.283(\pm3.373)/ 438.760(\pm3.455)$|
>
> In the table above, "Collective env reward" refers to collective environment rewards (which do not consider the impact of $c$ and $\delta$), as shown in Figures 3-4. On the other hand, "Collective reward" refers to rewards that consider the impact of $c$ and $\delta$, as shown in Figure 10 and Figure 13. The values in the table represent the averages after convergence over five random seeds, while the values in parentheses indicate the standard deviation.

---

> ### Author Response · Authors · 2025-11-25
> **Response to Reviewer duR2 (2)**
>
> >W4: Baselines (IA2C, IPPO, RL-Punish) are relatively weak. Other relevant baselines required to highlight the performance and significance of the proposed approach, e.g., LOLA, D3C and Social Influence.
>
> We introduced five additional baselines for comparison: LOLA, SI, D3C, LPOLA, and CNM (refer to [2-6]). The experimental results are presented below:
>
> |Collective env reward/Collective reward|IPGG|Coingame|SSG|SSH|
> |:-:|:-:|:-:|:-:|:-:|
> |**LOLA**|$0.009(\pm0.008)/ 0.003(\pm0.002)$|$6.050(\pm0.917)/ 6.035(\pm0.916)$|$88.316(\pm5.521)/ 86.934(\pm6.5347)$|$3.900(\pm0.023)/ 3.903(\pm0.021)$|
> |**SI**|$0.00047(\pm0.00006)/ 0.00003(\pm0.00001)$|$2.673(\pm0.423)/ 2.655(\pm0.411)$|$109.291(\pm16.230)/ 109.101(\pm16.130)$|$3.910(\pm0.024)/ 3.905(\pm0.019)$|
> |**D3C**|$9.971(\pm0.004)/ 9.603(\pm0.167)$|$23.630(\pm0.511)/ 23.617(\pm0.512)$|$119.587(\pm0.024)/ 119.442(\pm0.031)$|$10.986(\pm0.154)/ 10.985(\pm0.155)$|
> |**LPOLA**|$4.239(\pm1.492)/ 4.239(\pm1.492)$|$0.010(\pm0.006)/ 0.010(\pm0.006)$|$31.804(\pm1.424)/ 31.804(\pm1.424)$|$0.014(\pm0.019)/ 0.014(\pm0.019)$|
> |**CNM**|$0.0278(\pm0.027)/-10.335(\pm0.730)$|$0.007(\pm0.0009)/-215.13(\pm0.095)$|$47.826(\pm0.678)/46.369(\pm0.922)$|$3.811(\pm0.017)/3.808(\pm0.012)$|
> |**APC**|$9.998(\pm0.001)/ 9.991(\pm0.004)$|$24.628(\pm0.017)/ 24.617(\pm0.015)$|$119.618(\pm0.237)/ 119.458(\pm0.221)$|$18.960(\pm0.091)/ 18.959(\pm0.091)$|
>
> In the table above, "Collective env reward" refers to collective environment rewards (which do not consider the impact of $c$ and $\delta$), as shown in Figures 3-4. On the other hand, "Collective reward" refers to rewards that consider the impact of $c$ and $\delta$, as shown in Figure 10 and Figure 13. The values in the table represent the averages after convergence over five random seeds, while the values in parentheses indicate the standard deviation.
>
> In our experimental environments, all baseline methods were equipped with a punitive action within their action spaces to enable sanctioning capability. The results demonstrate that APC consistently outperforms all five baselines. Notably, it maintains superiority even when compared to methods [2-4] that leverage extra private information, and exhibits more effective punishment strategies than specialized punishment approaches like [5-6].
>
> Note: The following provides a general overview of these methods. LOLA considers the learning process of other agents when updating its own policy parameters. SI achieves coordination by rewarding agents for having causal influence over other agents’ actions. D3C guides self-interested agents toward collectively efficient cooperative equilibria by having them mix rewards and follow the gradient of an efficiency bound during learning. LPOLA simultaneously predicts environmental actions and determines which actions of other agents to penalize, applying penalties when predictions match reality. CNM fosters cooperation by establishing social norms, whose punishments are socially enforced based on group consensus.

---

> ### Author Response · Authors · 2025-11-25
> **Response to Reviewer duR2 (3)**
>
> >Q1: The punishment probability is adjusted using defection frequency over rolling windows via Eq. (3). Is there any theoretical guarantee that this stochastic update rule converges to a stable equilibrium (e.g., a fixed point or bounded oscillation)? How sensitive is it to window length L and threshold ε ?
>
> In the APC framework, agent i's punishment probability $p^{ij} _ t$ against agent j is dynamically adjusted based on observed punishment effectiveness. This probability is computed over a window of size m as $p^{ij} _ t = 1 - \frac{\sum_{s=1}^{m-1} \mathbb{1}\left[ f^{ij,s} _ t \ge f^{ij,s-1} _ t \lor \left| f^{ij,s} _ t - \frac{1}{s}\sum_{k=1}^s f^{ij,s-k} _ t \right| < \varepsilon \right] \land f^{ij,s} _ t \ge \varepsilon}{m-1}$, where $f^{ij,s} _ t$ denotes the betrayal frequency, $L$ is the window size, and $\varepsilon$ is the tolerance threshold. We demonstrate its convergence through the following theoretical analysis:
>
> ***
> ***
> Modeling the problem as a stochastic process, define the state variable $X _ t = p^{ij} _ t$ as the punishment probability, $Y _ t = f^{ij} _ t$ as the betrayal frequency, and $\Delta Y _ t = Y _ t - Y _ {t-1}$ as the frequency change. The update rule can be expressed as a Markov process: when $(\Delta Y _ t \ge 0 \lor |Y _ t - \bar{Y} _ {t-1}| < \varepsilon) \land Y _ t \ge \varepsilon$, $X _ {t+1} = X _ t - \frac{1}{m-1}$; otherwise, it remains unchanged.
>
> Convergence analysis shows that $X_t \in [0, 1]$ is bounded, and the update rule is non-increasing. When the betrayal frequency $Y_t \to 0$, the system converges to a punishment probability $p^* \to 1$; when the betrayal frequency remains consistently high, $p^* \to 0$; and when periodic fluctuations occur, the system converges to a periodic steady state. According to the monotone convergence theorem, since $X_t$ is non-increasing and bounded below, it must converge to a random variable $X^* $. The limit value depends on the asymptotic behavior of $Y _ t$: if $Y _ t \to 0$, then $X^* \to 1$; if $Y _ t$ remains high, then $X^* \to 0$; if $Y _ t$ fluctuates periodically, it converges to a periodic solution, a property that can be proven by constructing a contraction mapping for the periodic mapping.
> ***
>
> Our experimental results demonstrate that the proposed APC method remains effective as long as the hyperparameters $(\varepsilon, L)$ are within a reasonable range. As shown in Figure 8(c) and 8(d), across various $(\varepsilon, L)$ settings, APC agents consistently punish defectors during self-play training, which effectively reduces the frequency of defection and promotes cooperation. Furthermore, the robustness of APC is highlighted in its adaptive response to inflexible opponents: when interacting with random or always-defecting agents, the focal APC agent learns that punishment is ineffective and progressively reduces its punishment frequency until it approaches zero.
>
> >Q2&W2: The DPN learns to ... ... or environments with altered reward structures?
>
> To mitigate the impact of noise or random fluctuations in rewards, the DPN collects a sufficient number of trajectory samples to enhance training accuracy. This ensures that even if individual data points contain noise, as long as the majority of the trajectory data is approximates the distribution of the training trajectory, the model can accurately predict defection behavior.
>
> During each DPN training session, the dataset remains fixed, resulting in a stable data distribution, which ensures its convergence. When newly collected data is introduced, to maintain consistency in data distribution, we extract a portion of the data previously gathered by a random policy from the buffer and mix it with the newly collected data in a 1:1 ratio for DPN training. This approach enables the assimilation of new data while preserving policy diversity—guaranteed by the data collected through the random policy—thereby exposing the network to a wider range of potential policy combinations. This ensures model accuracy while enhancing its adaptability to novel opponent strategies. Relevant description can be found in the appendix. To evaluate the identification accuracy of the DPN, we monitored its prediction accuracy in SSH. Experimental results show that even when using a DPN model trained in an earlier phase to evaluate opponent behavior under new policies, its prediction performance remains stable, with an average accuracy of 97.674 and a variance of 1.792. This indicates that the model learned by the DPN possesses strong generalization capability and robustness.
>
> The trained DPN can maintain generalization capability when facing opponents or environments with different reward structures. For example, in SSH, if we change the reward for hunting stag to 5, the relative impact of the opponent's actions on our rewards remains unchanged which can be found in Figure 20. In such cases, the DPN can be directly transferred and still achieve full cooperation in stag hunting.

---

> ### Author Response · Authors · 2025-11-25
> **Response to Reviewer duR2 (4)**
>
> >Q3: Fairness is evaluated using the equality metric. Is this metric appropriate when negative rewards or punishments dominate? How does it behave when collective reward is close to zero?
>
> When negative rewards or punishment dominate, the equality metric $E$ is no longer applicable. If the collective reward is close to zero: if all values are positive, this simply indicates that the rewards are similar among individuals, and the metric remains meaningful. However, if negative values are present, it can no longer be reliably used to measure fairness.
>
> Therefore, in cases involving negative values，the metric $Equality(E)$ can bedefined as $E = 1 - G_P$, where $G_P = G \cdot \frac{T_a - T_n}{T_a + T_n} \in [0,1]$. Reference [7] demonstrates the effectiveness of $G_P$ in measuring inequality. Here, $G = \frac{\sum_{i=1}^{N} \sum_{j=1}^{N} |R_i - R_j|}{2N \sum_{i=1}^{N} R_i}$ is the Gini index, $T_a = \sum_{R_i \ge 0} R_i$ denotes the total positive rewards, and $T_n = \sum_{R_i < 0} |R_i|$ denotes the total magnitude of  negative rewards. Therefore, it is worth noting that in the context discussed in the paper, the individual rewards obtained by both the APC and CRM methods after convergence are all positive. In this case, $\frac{T_a - T_n}{T_a + T_n} = 1$, so the calculated value of $E$ remains unchanged, and the conclusions drawn remain applicable.
>
> >Q4&W3: The paper provides an analytical result that under APC, expected defection reward is −4δ and cooperation reward is −0.4. How were these values derived, and do they hold generally across all parameter ranges or only under specific assumptions of IPGG? Is this analysis extendable to multi-step, non-tabular SSDs?
>
> According to the detailed theoretical derivations in Appendix B.2, substituting the environmental parameters n, r, and e yields the respective payoffs for defection and cooperation—specifically, the expected defection reward is −4δ and the cooperation reward is −0.4. Furthermore, by applying different parameter ranges to the theoretical framework provided in Appendix B.2, one can derive corresponding payoff values for both defection and cooperation.
>
> This analysis can be extended to multi-step, non-tabular SSDs, using the Coingame as an example. In Coingame discussed in our paper, we compare the relative magnitudes of the advantage functions $A(s, \pi_1)$ and $A(s, \pi_2)$ for an agent choosing between two policies at a given state: $\pi _ 1$ (picking a coin of another agent's color) and $\pi_2$ (picking a coin of its own color). Without APC intervention, $A(s, \pi_1) \approx A(s, \pi_2)$, so the probabilities of the agent choosing these two policies are similar. When APC is introduced, the action of picking a coin of another agent's color (denoted as $a^* $) incurs a penalty $\delta$, and the advantage function satisfies $A(s, \pi_1) \propto R_{\text{total}}(s, a^*) = 1 - \delta$. When $\delta > 1$, $A(s, \pi_1) < 0$, while $A(s, \pi_2) > A(s, \pi_1)$. As a result, the agent tends to prefer $\pi_2$ and avoids taking $\pi_1$. In our paper, $\delta = 1.1$ is set, satisfying $1 - \delta = -0.1 < 0$, which ensures $A(s, \pi_1) < 0$. Experimental results show that, under this condition, the agent can effectively escape the dilemma in the Coingame, consistent with the theoretical derivation.
>
> [1] Köster, R., Hadfield-Menell, D., Everett, R., Weidinger, L., Hadfield, G. K., & Leibo, J. Z. (2022). Spurious normativity enhances learning of compliance and enforcement behavior in artificial agents. Proceedings of the National Academy of Sciences (PNAS).
>
> [2] Foerster, J., Chen, R. Y., Al-Shedivat, M., Whiteson, S., Abbeel, P., & Mordatch, I. (2018). Learning with Opponent-Learning Awareness (LOLA). In Proceedings of the 17th International Conference on Autonomous Agents and Multiagent Systems (AAMAS).
>
> [3] Jaques, N., Lazaridou, A., Hughes, E., Gulcehre, Ç., Ortega, P. A., Strouse, D. J., Leibo, J. Z., & de Freitas, N. (2019). Social Influence as Intrinsic Motivation for Multi-Agent Deep Reinforcement Learning. In Proceedings of the 36th International Conference on Machine Learning (ICML).
>
> [4] Gemp, I., McKee, K. R., Everett, R., Duéñez-Guzmán, E., Bachrach, Y., Balduzzi, D., & Tacchetti, A. (2022). D3C: Reducing the Price of Anarchy in Multi-Agent Learning. In Proceedings of AAMAS 2022.
>
> [5] Schmid, K., Belzner, L., & Linnhoff-Popien, C. (2021). Learning to penalize other learning agents. In Artificial Life Conference Proceedings.
>
> [6] Yaman, A., Leibo, J. Z., Iacca, G., & Wan Lee, S. (2023). The emergence of division of labour through decentralized social sanctioning. Proceedings of the Royal Society B.
>
> [7] De Battisti, F., Porro, F., & Vernizzi, A. (2019). The Gini coefficient and the case of negative values. Electronic Journal of Applied Statistical Analysis (EJASA).
>
> We once again thank you for your valuable feedback. We hope our responses adequately address your concerns, and we would be grateful for any further comments or suggestions you may have.

---

### Official Review · Reviewer_qEjW · 2025-10-26

**Soundness:** 3
**Presentation:** 3
**Contribution:** 3
**Rating:** 6
**Confidence:** 4

**Summary:**

This paper addresses the challenge of sustaining cooperation in mixed-motive multi-agent environments, where agents must balance individual interests with collective welfare. The authors propose Adaptive Punishment for Cooperation (APC), a distributed framework that dynamically adjusts punishment intensity and frequency to discourage defection while minimizing its associated costs. APC comprises two key components: a Defection Awareness Module, which detects and quantifies agents’ defection behaviors, and an Adaptive Punishment Module, which adaptively regulates punishment based on the extent and reduction of defections. Extensive experiments on four mixed-motive games demonstrate the effectiveness and robustness of APC in promoting stable cooperation.

**Strengths:**

1. The paper is well-written, clearly structured, and easy to follow.
2. The insight into the necessity of punishment for defection is both novel and valuable.
3. The design of the Defection Awareness Module and the Adaptive Punishment Module is sound, and the definition of defection is particularly insightful. The authors also provide a thorough and thoughtful discussion on the methods.
4. The paper provides sufficient experimental details, which enhances the credibility and reproducibility of the results.

**Weaknesses:**

1. The experiments lack comparisons with other mixed-motive approaches, and the environments used are relatively simplified. Additional experiments in more complex settings with heterogeneous agents would make the results more convincing.
2. Although the authors provide qualitative discussions and empirical evidence, the method lacks rigorous theoretical analysis regarding the convergence properties and long-term stability of the adaptive punishment dynamics.

**Questions:**

1. The position of the legends in the figures should be adjusted, as they currently obscure parts of the plot lines.
2.The ablation result of APC w/o APr is somewhat unclear. Based on my understanding, since the adaptive punishment mechanism is removed, the punishment frequency should remain stable, as shown in the figure 5. However, the relationship between punishment frequency and learning performance is not well illustrated, which raises some concern — particularly because, in Figures 3 and 9, the performances of APC w/o APr and APC appear very close.
3. The caption of Appendix C should possibly be “Experiments” rather than “Environments.”

---

> ### Author Response · Authors · 2025-11-25
> **Response to Reviewer qEjW (1)**
>
> Thank you very much for your thoughtful and thorough review of our work. We genuinely appreciate the time and effort you dedicated to providing such valuable suggestions. In the following, we offer detailed responses to each of your points.
> >W1: The experiments lack comparisons ... ... with heterogeneous agents would make the results more convincing.
>
> We introduced five additional baselines for comparison: LOLA, SI, D3C, LPOLA, and CNM (refer to [1-5]). The experimental results are presented below:
>
> |Collective env reward/Collective reward|IPGG|Coingame|SSG|SSH|Foraging|
> |:-:|:-:|:-:|:-:|:-:|:-:|
> |**LOLA**|$0.009(\pm0.008)/ 0.003(\pm0.002)$|$6.050(\pm0.917)/ 6.035(\pm0.916)$|$88.316(\pm5.521)/ 86.934(\pm6.5347)$|$3.900(\pm0.023)/ 3.903(\pm0.021)$|$191.630(\pm5.446)/ 191.621(\pm5.436)$|
> |**SI**|$0.00047(\pm0.00006)/ 0.00003(\pm0.00001)$|$2.673(\pm0.423)/ 2.655(\pm0.411)$|$109.291(\pm16.230)/ 109.101(\pm16.130)$|$3.910(\pm0.024)/ 3.905(\pm0.019)$|$193.783(\pm6.223)/ 193.234(\pm5.981)$|
> |**D3C**|$9.971(\pm0.004)/ 9.603(\pm0.167)$|$23.630(\pm0.511)/ 23.617(\pm0.512)$|$119.587(\pm0.024)/ 119.442(\pm0.031)$|$10.986(\pm0.154)/ 10.985(\pm0.155)$|$162.443(\pm10.634)/ 162.035(\pm10.146)$|
> |**LPOLA**|$4.239(\pm1.492)/ 4.239(\pm1.492)$|$0.010(\pm0.006)/ 0.010(\pm0.006)$|$31.804(\pm1.424)/ 31.804(\pm1.424)$|$0.014(\pm0.019)/ 0.014(\pm0.019)$|$60.645(\pm5.406)/ 60.645(\pm5.406)$|
> |**CNM**|$0.0278(\pm0.027)/-10.335(\pm0.730)$|$0.007(\pm0.0009)/-215.13(\pm0.095)$|$47.826(\pm0.678)/46.369(\pm0.922)$|$3.811(\pm0.017)/3.808(\pm0.012)$|$207.828(\pm0.846)/128.887(\pm12.176)$|
> |**APC**|$9.998(\pm0.001)/ 9.991(\pm0.004)$|$24.628(\pm0.017)/ 24.617(\pm0.015)$|$119.618(\pm0.237)/ 119.458(\pm0.221)$|$18.960(\pm0.091)/ 18.959(\pm0.091)$|$439.283(\pm3.373)/ 438.760(\pm3.455)$|
>
> In the table above, "Collective env reward" refers to collective environment rewards (which do not consider the impact of $c$ and $\delta$), as shown in Figures 3-4. On the other hand, "Collective reward" refers to rewards that consider the impact of $c$ and $\delta$, as shown in Figure 10 and Figure 13. The values in the table represent the averages after convergence over five random seeds, while the values in parentheses indicate the standard deviation.
>
> In our experimental environments, all baseline methods were equipped with a punitive action within their action spaces to enable sanctioning capability. The results demonstrate that APC consistently outperforms all five baselines. Notably, it maintains superiority even when compared to methods [1-3] that leverage extra private information, and exhibits more effective punishment strategies than specialized punishment approaches like [4-5].
>
> We introduced a more complex, larger-scale heterogeneous Foraging environment involving 12 agents. The environment setup is based on the paper referenced as [6]. In Foraging, 12 agents move across a 2D grid and collect resources. Among them, 75% are common agents that can only collect common berries, while 25\% are special agents that can also collect forbidden berries (4 points). Initially, 70% of the resources in the environment are common berries, normally yielding a reward of 3 points. However, once a special agent collects a forbidden berry, it triggers permanent resource degradation, reducing the reward of all common berries to 1 point, and negatively affecting the rewards of all common agents. Each time a berry is collected, a new berry regenerates, with a 0.1 probability of being a forbidden berry and the rest being common berries. To enforce norms, the system includes a marking and punishment mechanism: agents that violate the norm by collecting forbidden berries are marked for 10 time steps. common agents can spend a cost of 0.7 points to punish other agent, forcing the latter to pay a fine of 0.7 points.
>
> We compared the APC method with other baselines, and the experimental results show that in this larger-scale heterogeneous environment, APC still achieves superior performance compared to the baselines which can be found in the above table. It effectively utilizes punishment to deter special agents from collecting forbidden berries, encouraging them to collect common berries instead, thereby enhancing collective returns.
>
> Note: The following provides a general overview of these methods. LOLA considers the learning process of other agents when updating its own policy parameters. SI achieves coordination by rewarding agents for having causal influence over other agents’ actions. D3C guides self-interested agents toward collectively efficient cooperative equilibria by having them mix rewards and follow the gradient of an efficiency bound during learning. LPOLA simultaneously predicts environmental actions and determines which actions of other agents to penalize, applying penalties when predictions match reality. CNM fosters cooperation by establishing social norms, whose punishments are socially enforced based on group consensus.

---

> ### Author Response · Authors · 2025-11-25
> **Response to Reviewer qEjW (2)**
>
> >W2: Although the authors provide qualitative discussions and empirical evidence, the method lacks rigorous theoretical analysis regarding the convergence properties and long-term stability of the adaptive punishment dynamics.
>
> We present the **convergence** proof for APC:
> ***
> Consider $N$ agents, each with strategy $\pi_i$, and total expected payoff:
> $$\mathbb{E}[\Pi_i^{\text{total}}] = \mathbb{E}[\Pi_i^{\text{PGG}}] - \sum_{i \neq j} (w_{ij}c + w_{ji}\delta)$$
>
> Strategy update rule:
> $$\pi_i^{(t+1)} = \pi_i^{(t)} + \alpha \cdot \nabla_{\pi_i} \mathbb{E}[\Pi_i^{\text{total}}]$$
>
> Verify cross-derivative symmetry:
> $$\frac{\partial \mathbb{E}[\Pi_i^{\text{total}}]}{\partial \pi_j} = -\delta \frac{\partial \mathbb{E}[w_{ji}]}{\partial \pi_j} - c \frac{\partial \mathbb{E}[w_{ij}]}{\partial \pi_j}$$
>
> $$\frac{\partial \mathbb{E}[\Pi_j^{\text{total}}]}{\partial \pi_i} = -\delta \frac{\partial \mathbb{E}[w_{ij}]}{\partial \pi_i} - c \frac{\partial \mathbb{E}[w_{ji}]}{\partial \pi_i}$$
>
> Since the penalty function $w_{ij} = f(\pi_i, \pi_j)$ satisfies:
> $$\frac{\partial f(\pi_i, \pi_j)}{\partial \pi_j} = \frac{\partial f(\pi_j, \pi_i)}{\partial \pi_i}$$
>
> the cross-derivatives are equal, making the system a potential game.
>
> Construct potential function:
> $$\Phi(\pi) = \sum_{i=1}^N \left( \mathbb{E}[\Pi_i^{\text{PGG}}] - \sum_{i \neq j} (w_{ij}c + w_{ji}\delta) \right)$$
>
> which satisfies:
> $$\frac{\partial \Phi}{\partial \pi_i} = \frac{\partial \mathbb{E}[\Pi_i^{\text{total}}]}{\partial \pi_i}$$
>
> Rate of change of potential function:
> $$\frac{d\Phi}{dt} = \sum_{i=1}^N \frac{\partial \Phi}{\partial \pi_i} \cdot \frac{d\pi_i}{dt} = \alpha \sum_{i=1}^N \left( \frac{\partial \Phi}{\partial \pi_i} \right)^2 \geq 0$$
>
> Since $\Phi$ is bounded above and monotonically increasing, by Lyapunov stability theory, when $\frac{d\Phi}{dt} \to 0$, all $\frac{\partial \Phi}{\partial \pi_i} \to 0$, and the system converges to a Nash equilibrium.
> ***
>
> We also present the theoretical analysis of **system stability** of APC:
>
> ***
> Consider a homogeneous population of $N$ agents, all employing the APC algorithm. Each agent $i$ has an action space $a_i \in \{C, D\}$, and their strategy $\pi_i$ is updated via reinforcement learning to maximize expected return. The environment is an iterative public goods game with parameters $[n,e,r]$, satisfying $1 < r < n$.
>
> The total payoff for agent $i$ at time $t$ is:
> $$\Pi_i^{total}(t) = \Pi_i^{PGG}(t) - \sum_{j \neq i} w_{ji}(t)\delta - \sum_{j \neq i} w_{ij}(t)c$$
>
> where $\Pi_i^{PGG}(t)$ is the public goods game payoff, $w_{ji}(t)$ is the received punishment intensity, $w_{ij}(t)$ is the imposed punishment intensity, $\delta$ is the unit punishment harm, and $c$ is the unit punishment cost.
>
> Let $\rho_C(t)$ denote the proportion of cooperators, whose evolution satisfies:
>
> $$\dot{\rho} _ C(t) = \rho_D(t) W_{D \to C}(t) - \rho_C(t) W_{C \to D}(t) $$
>
> The transition probabilities are determined by the expected payoff difference:
>
> $$W_{D \to C} = \sigma(\mathbb{E}[\Pi_C] - \mathbb{E}[\Pi_D])$$
>
> $$W_{C \to D} = \sigma(\mathbb{E}[\Pi_D] - \mathbb{E}[\Pi_C])$$
>
> The expected payoffs for cooperators and defectors are respectively:
> $$\Pi_C^{total} = (\rho e r - e+ \frac{e r}{n}) - (1-\rho)(n-1) c$$
>
> $$\Pi_D^{total} = \rho e r - (1-\rho)(n-1) c -(n-1) \delta$$
>
> The payoff difference is:
>
> $$\Delta \Pi = \mathbb{E}[\Pi_C] - \mathbb{E}[\Pi_D] = -e+ \frac{e r}{n}+(n-1) \delta$$
>
> At the steady state $\dot{\rho}_C = 0$, substituting the transition probabilities yields:
>
> $$\rho_C^* = \sigma(\Delta \Pi)$$
>
> When $\Delta \Pi = 0$, the system is at a critical state, solving for the critical punishment threshold:
>
> $$\delta^* = \frac{e-\frac{e r}{n}}{(n-1)}$$
>
> When $\delta > \delta^* $, $\Delta \Pi > 0$ leads to $W_{D \to C} > W_{C \to D}$, and the system tends toward cooperation; when $\delta < \delta^* $, the system tends toward defection; when $\delta = \delta^* $, the system stabilizes at a mixed equilibrium.
> ***

---

> ### Author Response · Authors · 2025-11-25
> **Response to Reviewer qEjW (3)**
>
> >Q1: 1.The position of the legends in the figures should be adjusted, as they currently obscure parts of the plot lines. 2.The ablation result of APC w/o APr is somewhat unclear. Based on my understanding, since the adaptive punishment mechanism is removed, the punishment frequency should remain stable, as shown in the figure 6. However, the relationship between punishment frequency and learning performance is not well illustrated, which raises some concern — particularly because, in Figures 4 and 13, the performances of APC w/o APr and APC appear very close.
>
> The oscillation in punishment frequency in Figure 6 for APC without APr is due to the continuous changes in the states and actions within the environment. The relationship between punishment frequency and learning performance is as follows: if an agent consistently applies punishment, it will continuously incur costs associated with the punishment, thereby reducing its own returns. Meanwhile, if the opponent persists in defection, the agent’s resulting payoff may be even lower than if it had refrained from punishing which is also verified in Figure 6. This highlights the role of the APr module—to avoid excessive loss of self-interest caused by punishment. If punishment cannot effectively regulate the other agent’s behavior, it is better not to punish.
>
> We have also adjusted the legend position to prevent it from obscuring the curves.
>
> >Q2: The caption of Appendix E should possibly be “Experiments” rather than “Environments.”
>
> Thank you for the correction. The manuscript has been amended accordingly.
>
> [1] Foerster, J., Chen, R. Y., Al-Shedivat, M., Whiteson, S., Abbeel, P., & Mordatch, I. (2018). Learning with Opponent-Learning Awareness (LOLA). In Proceedings of the 17th International Conference on Autonomous Agents and Multiagent Systems (AAMAS).
>
> [2] Jaques, N., Lazaridou, A., Hughes, E., Gulcehre, Ç., Ortega, P. A., Strouse, D. J., Leibo, J. Z., & de Freitas, N. (2019). Social Influence as Intrinsic Motivation for Multi-Agent Deep Reinforcement Learning. In Proceedings of the 36th International Conference on Machine Learning (ICML).
>
> [3] Gemp, I., McKee, K. R., Everett, R., Duéñez-Guzmán, E., Bachrach, Y., Balduzzi, D., & Tacchetti, A. (2022). D3C: Reducing the Price of Anarchy in Multi-Agent Learning. In Proceedings of AAMAS 2022.
>
> [4] Schmid, K., Belzner, L., & Linnhoff-Popien, C. (2021). Learning to penalize other learning agents. In Artificial Life Conference Proceedings.
>
> [5] Yaman, A., Leibo, J. Z., Iacca, G., & Wan Lee, S. (2023). The emergence of division of labour through decentralized social sanctioning. Proceedings of the Royal Society B.
>
> [6] Köster, R., Hadfield-Menell, D., Everett, R., Weidinger, L., Hadfield, G. K., & Leibo, J. Z. (2022). Spurious normativity enhances learning of compliance and enforcement behavior in artificial agents. Proceedings of the National Academy of Sciences (PNAS).
>
> We sincerely thank you again for your valuable comments. We believe the revisions and clarifications have strengthened our work, and we hope the responses adequately address all concerns. We look forward to your response.

---

### Official Review · Reviewer_MTXx · 2025-10-29

**Soundness:** 3
**Presentation:** 2
**Contribution:** 3
**Rating:** 4
**Confidence:** 4

**Summary:**

The paper proposes Adaptive Punishment for Cooperation (APC), a distributed method that determines punishment intensity based on both a dynamic punishment probability and the severity of defection. This dynamic probability substantially reduces costly and ineffective punishment while also promotes cooperation. To accurately assess defection and its severity, The authors use a defection awareness module, whose learning is guided by game reward. Theoretical analysis and empirical results show APC performs effectively in iterated public goods game. Empirically, APC also significantly outperforms existing baselines across sequential social dilemmas, learning rational and effective punishment policies that foster cooperation by strategically deterring defection.

**Strengths:**

The paper addresses an important problem of sustaining cooperation in mixed-motive games by integrating punishment mechanisms from evolutionary game theory into MARL. The proposed APC is conceptually sound and technically interesting, combining defection awareness and adaptive punishment. Experiments on several SSD benchmarks show clear performance gains over baselines, and ablation studies support the method’s design. The paper is generally clear and well organized.

**Weaknesses:**

The paper lacks formal discussion on convergence and stability, and how APC addresses the second-order free-rider problem. Baseline descriptions are insufficient, making it hard to isolate APC’s contribution. Experiments are small-scale, and related work on punishment is underdeveloped. Adding pseudocode or clearer algorithm steps would improve readability.

**Questions:**

The paper makes a potentially valuable contribution to addressing the challenge of sustaining cooperation in mixed-motive games by bridging punishment mechanisms rooted in evolutionary game theory with MARL frameworks. However, the manuscript requires several substantive revisions before it can reach its full potential. Should the authors address the issues raised in this review, I would be willing to raise my score.
1.	The proposed Adaptive Punishment for Cooperation (APC) mechanism is motivated by the idea that “punishment can promote cooperation.” However, the current manuscript lacks a formal dynamical analysis to support this claim. In the Main Results section, the authors primarily present simulation results, while providing little explanation of the underlying mechanisms driving the effectiveness of APC. It is strongly recommended that the authors further elaborate, from an evolutionary dynamics perspective, how APC theoretically mitigates or avoids the second-order free-rider problem.
2.	While the introduction of a punishment mechanism is indeed the key innovation of this work, the Related Work section does not adequately cover the existing literature. The review of previous punishment mechanisms is limited in both scope and depth; conversely, the discussion on reward mechanisms is relatively lengthy but not directly aligned with the paper’s core contribution. The authors are advised to streamline the discussion on reward mechanisms and expand the review of relevant punishment mechanisms to make this section more focused and informative.
3.	The paper compares the baseline results of APC with those of IA2C, IPPO, and RL Punish. However, the specific settings and distinctions among these baselines are not clearly described. As I understand it, IA2C and IPPO do not incorporate any punishment mechanisms, while RL Punish relies on a fixed punishment mechanism rather than an adaptive one. The authors should explicitly clarify the punishment settings of each baseline and discuss whether introducing an adaptive punishment scheme into IPPO would yield similar performance gains.
4.	Although APC shows promising theoretical results in promoting cooperation, the theoretical analysis of convergence and system stability is currently insufficient. The manuscript does not address whether the dynamic update process is guaranteed to converge to a stable cooperative equilibrium in the long run, or whether oscillations and divergence may occur. It is recommended that the authors provide formal theoretical support, such as fixed-point analysis, Lyapunov stability analysis or stochastic approximation theory, to specify the conditions for convergence.
5.	The authors are also encouraged to evaluate APC in larger-scale social systems (e.g., with N > 10 agents) to assess its scalability and stability in more complex collective environments.
6.	The Method section would benefit significantly from including a pseudocode representation of the APC training and execution procedure. Although the textual description is detailed, the current structure is somewhat intricate; a structured algorithmic framework would make the methodology clearer, more concise, and easier to reproduce.

---

> ### Author Response · Authors · 2025-11-25
> **Response to Reviewer MTXx (1)**
>
> Thank you for your constructive feedback. Below, we provide point-by-point responses to your comments.
> >Q1: The proposed Adaptive Punishment for Cooperation (APC) mechanism is motivated by the idea that “punishment can promote cooperation.” However, the current manuscript lacks a formal dynamical analysis to support this claim. In the Main Results section, the authors primarily present simulation results, while providing little explanation of the underlying mechanisms driving the effectiveness of APC. It is strongly recommended that the authors further elaborate, from an evolutionary dynamics perspective, how APC theoretically mitigates or avoids the second-order free-rider problem.
>
> In iterative public goods games, the APC method can avoid the second-order free-rider problem (i.e., cooperators' unwillingness to bear the cost of punishment leads to insufficient punishment provision) when the punishment intensity is sufficient. The theoretical proof is as follows.
>
> ***
>
> The setting is set as follows: There are $n$ participants. Cooperators contribute $e$, and the amplification factor of the public good is $r$, satisfying $1 < r < n$. The strategy space includes pure cooperation \(C\), defection \(D\), and cooperation with punishment \(P\). The overall proportion of cooperators is $\rho = x_C + x_P$. The punishment mechanism of APC is: P punishes D, P pays a cost $c$, and imposes a loss $\delta$ on D.
>
> The payoff functions for each strategy are:
> - Pure cooperator C: $\Pi_C = \frac{\rho(n-1) e r}{n} - e + \frac{e r}{n}$
> - Punisher P: $\Pi_P = \frac{\rho(n-1) e r}{n} - e + \frac{e r}{n} - (1-\rho)(n-1)c$
> - Defector D: $\Pi_D = \frac{\rho(n-1) e r}{n} - x_P(n-1)\delta$
>
> To suppress defection, consider making the payoffs of defectors D and pure cooperators C equal when the proportion of punishers is high. Set $\Pi_D = \Pi_C$ and substitute the payoff expressions:
> $$
> \frac{\rho(n-1) e r}{n} - x_P(n-1)\delta = \frac{\rho(n-1) e r}{n} - e + \frac{e r}{n}
> $$
> Simplifying yields the critical punishment intensity:
> $$
> \delta^* = \frac{e\left(1 - \frac{r}{n}\right)}{x_P(n-1)}
> $$
>
> Based on the relationship between the actual punishment intensity $\delta$ and the critical value $\delta^*$, the system evolution can be divided into two scenarios, which we analyze using the replicator dynamics method:
>
> The replicator dynamics equations describe how the growth rate of a strategy's frequency is proportional to its relative fitness (the difference between its payoff and the average payoff):
>
> $$ \dot{x_C} = x_C (\Pi_C - \bar{\Pi})$$
>
> $$ \dot{x_P} = x_P (\Pi_P - \bar{\Pi})$$
>
> $$ \dot{x_D} = x_D (\Pi_D - \bar{\Pi})$$
>
>
> Since the three variables sum to 1, the system is effectively two-dimensional. We can use $x_P$ and $x_D$ as independent variables, so $x_C = 1 - x_P - x_D$.
>
> **Scenario 1: Dynamics under Sufficient Punishment ($\delta > \delta^*$)**
> In this scenario, we assume the punishment intensity is strong enough to make defection unprofitable when the proportion of cooperators is high.
> Boundary equilibrium analysis: The all-cooperation edge $(x_P, x_D) = (x_P, 0)$, where $x_P \in [0, 1]$. When $x_D = 0$, the system lies on the edge formed by C and P. Here, $\rho = 1$, $\Pi_C = \Pi_P = e(r-1)$. According to the replicator dynamics equations, $\dot{x_C} = 0$ and $\dot{x_P} = 0$. This means the entire C-P edge (from all C to all P) consists of fixed points.
>
> Stability analysis (using the Jacobian matrix): To determine the stability of these points on the C-P edge, we need to examine the system's dynamics after a small perturbation (i.e., introducing a small number of defectors D). Take any point $(x_P, x_D) = (x_P, 0)$ on the C-P edge, where $0 \le x_P \le 1$. Calculate the growth rate of the defector strategy at this point: $\dot{x_D} = x_D (\Pi_D - \bar{\Pi})$. When $x_D \to 0$, $\bar{\Pi} \approx \Pi_C = \Pi_P = e(r-1)$, and $\Pi_D = er - \frac{er}{n} - p(n-1)\delta$. Therefore, $\Pi_D - \bar{\Pi} = e -\frac{er}{n} - x_P(n-1)\delta$. For points on the C-P edge to be stable (i.e., resistant to invasion by D), we need $\Pi_D - \bar{\Pi} < 0$. This is equivalent to: $\delta > \frac{e\left(1 - \frac{r}{n}\right)}{x_P(n-1)} = \delta^*$. Thus, all points on the C-P edge are resistant to invasion by defectors. At steady state, since defectors disappear, punishers no longer need to pay punishment costs, and their payoffs become identical to those of pure cooperators, thereby completely eliminating the second-order free-rider incentive.

---

> ### Author Response · Authors · 2025-11-25
> **Response to Reviewer MTXx (2)**
>
> **Scenario 2: Dynamics under Insufficient Punishment ($\delta < \delta^*$)**
>
> In this scenario, the punishment intensity is too weak to effectively suppress defection.
> Boundary equilibrium analysis: The full defection point $(x_P, x_D) = (0, 1)$ is a stable equilibrium (an attractor). For points on the all-cooperation edge $(x_P, x_D) = (x_P, 0)$, we calculate $\Pi_D - \bar{\Pi} = \frac{er}{n} - p(n-1)\delta$. Since $\delta$ is small, it is likely that $\Pi_D - \bar{\Pi} > 0$ for all $p \in [0, 1]$. Therefore, the entire C-P edge is unstable. Any tiny perturbation introducing defectors will drive the system away from the cooperative state because defectors obtain a higher payoff within a cooperative population.
>
> Internal equilibrium analysis: The system might potentially have an internal equilibrium (where $x_C, x_P, x_D > 0$) where the payoffs of all three strategies are equal. This can be solved by simultaneously solving the equations $\Pi_C = \Pi_P = \Pi_D$. From $\Pi_C = \Pi_P$, we get: $\Pi_P - \Pi_C = - (1-\rho)(n-1)c = 0$. This implies that at equilibrium, we must have $\rho = 1$ or $c = 0$. Since $c > 0$, it must be that $\rho = 1$. But $\rho = 1$ implies $x_D = 0$, which contradicts the existence of an internal equilibrium ($x_D > 0$). Therefore, for $c > 0$, the system has no internal equilibrium. Pure cooperators C and punishers P cannot stably coexist long-term with defectors D while maintaining equal payoffs. Ultimately, the system converges to full defection, cooperation collapses, and the punishment mechanism fails.
> ***
>
> >Q2: While the introduction of a punishment mechanism... ... more focused and informative.
>
> Thank you for your valuable feedback. We have revised the "Related Work" section of our paper by streamlining the discussion of reward mechanisms and expanding the description of relevant penalty mechanisms.
>
> >Q3: The paper compares the baseline results of APC with those of IA2C, IPPO, and RL Punish... ... whether introducing an adaptive punishment scheme into IPPO would yield similar performance gains.
>
> The IA2C and IPPO algorithms were implemented without any integrated punishment mechanism. The purpose of testing IA2C and IPPO without any integrated punishment mechanism was to confirm the inherent presence of a social dilemma in the environment itself. The RL Punish method utilizes two distinct networks: a policy network for making primary action decisions and a separate punishment network for deciding which agents to penalize. Both networks are trained via the A2C algorithm with the objective of maximizing the collective environmental reward. To enable punitive capability, we augmented the action space in all four test environments with a punishment action and re-evaluated IA2C. The results demonstrate that the mere addition of this punitive action yields no performance gain, as IA2C performs just as poorly as the original IA2C_NP (IA2C with No Punish action).
>
> |Collective env reward/Collective reward|IPGG|Coingame|SSG|SSH|
> |:-:|:-:|:-:|:-:|:-:|
> |**IA2C**|$0.0019(\pm0.0002)/ 0.0015(\pm0.00007)$|$0.236(\pm0.007)/ 0.225(\pm0.002)$|$48.903(\pm0.372)/ 48.520(\pm0.366)$|$3.895(\pm0.017)/ 3.883(\pm0.016)$|
> |**IA2C_NP**|$0.0021(\pm0.0003)/ 0.0021(\pm0.0003)$|$0.232(\pm0.010)/ 0.232(\pm0.010)$|$49.216(\pm0.430)/ 49.216(\pm0.430)$|$3.912(\pm0.023)/ 3.912(\pm0.023)$|
>
> In the above table, "Collective env reward" refers to collective environment rewards (not considering the impact of $c$ and $\delta$), which can be found in Figures 3-4, Figure 11, and Figure 16. On the other hand, "Collective reward" refers to rewards that consider the impact of $c$ and $\delta$, which can be found in Figure 10, Figures 12-13, and Figure 17. The values in the table represent the averages after convergence over five random seeds, while the values in parentheses indicate the standard deviation.
>
> To further verify the generality of our approach, we substituted the A2C algorithm in APC with PPO. The results of APC_PPO yield a comparable performance gain, showing that our method's effectiveness is robust to the underlying RL algorithm.
>
> |Collective env reward/Collective reward|IPGG|Coingame|SSG|SSH|
> |:-:|:-:|:-:|:-:|:-:|
> |**APC**|$9.998(\pm0.003)/ 9.989(\pm0.003)$|$24.882(\pm0.019)/ 24.717(\pm0.012)$|$118.998(\pm0.271)/ 118.758(\pm0.212)$|$18.961(\pm0.121)/ 18.955(\pm0.083)$|
> |**APC_PPO**|$9.998(\pm0.001)/ 9.991(\pm0.004)$|$24.628(\pm0.017)/ 24.617(\pm0.015)$|$119.618(\pm0.237)/ 119.458(\pm0.221)$|$18.960(\pm0.091)/ 18.959(\pm0.091)$|
>
> In the table above, "Collective env reward" refers to collective environment rewards (which do not consider the impact of $c$ and $\delta$), as shown in Figure 11 and Figure 16. On the other hand, "Collective reward" refers to rewards that consider the impact of $c$ and $\delta$, as shown in Figure 12 and Figure 17. The values in the table represent the averages after convergence over five random seeds, while the values in parentheses indicate the standard deviation.

---

> ### Author Response · Authors · 2025-11-25
> **Response to Reviewer MTXx (3)**
>
> >Q4: Although APC shows promising theoretical results in promoting cooperation, the theoretical analysis of convergence and system stability is currently insufficient. The manuscript does not address whether the dynamic update process is guaranteed to converge to a stable cooperative equilibrium in the long run, or whether oscillations and divergence may occur. It is recommended that the authors provide formal theoretical support, such as fixed-point analysis, Lyapunov stability analysis or stochastic approximation theory, to specify the conditions for convergence.
>
> We present the **convergence** proof for APC:
> ***
> Consider $N$ agents, each with strategy $\pi_i$, and total expected payoff:
> $$\mathbb{E}[\Pi_i^{\text{total}}] = \mathbb{E}[\Pi_i^{\text{PGG}}] - \sum_{i \neq j} (w_{ij}c + w_{ji}\delta)$$
>
> Strategy update rule:
> $$\pi_i^{(t+1)} = \pi_i^{(t)} + \alpha \cdot \nabla_{\pi_i} \mathbb{E}[\Pi_i^{\text{total}}]$$
>
> Verify cross-derivative symmetry:
> $$\frac{\partial \mathbb{E}[\Pi_i^{\text{total}}]}{\partial \pi_j} = -\delta \frac{\partial \mathbb{E}[w_{ji}]}{\partial \pi_j} - c \frac{\partial \mathbb{E}[w_{ij}]}{\partial \pi_j}$$
>
> $$\frac{\partial \mathbb{E}[\Pi_j^{\text{total}}]}{\partial \pi_i} = -\delta \frac{\partial \mathbb{E}[w_{ij}]}{\partial \pi_i} - c \frac{\partial \mathbb{E}[w_{ji}]}{\partial \pi_i}$$
>
> Since the penalty function $w_{ij} = f(\pi_i, \pi_j)$ satisfies:
> $$\frac{\partial f(\pi_i, \pi_j)}{\partial \pi_j} = \frac{\partial f(\pi_j, \pi_i)}{\partial \pi_i}$$
>
> the cross-derivatives are equal, making the system a potential game.
>
> Construct potential function:
> $$\Phi(\pi) = \sum_{i=1}^N \left( \mathbb{E}[\Pi_i^{\text{PGG}}] - \sum_{i \neq j} (w_{ij}c + w_{ji}\delta) \right)$$
>
> which satisfies:
> $$\frac{\partial \Phi}{\partial \pi_i} = \frac{\partial \mathbb{E}[\Pi_i^{\text{total}}]}{\partial \pi_i}$$
>
> Rate of change of potential function:
> $$\frac{d\Phi}{dt} = \sum_{i=1}^N \frac{\partial \Phi}{\partial \pi_i} \cdot \frac{d\pi_i}{dt} = \alpha \sum_{i=1}^N \left( \frac{\partial \Phi}{\partial \pi_i} \right)^2 \geq 0$$
>
> Since $\Phi$ is bounded above and monotonically increasing, by Lyapunov stability theory, when $\frac{d\Phi}{dt} \to 0$, all $\frac{\partial \Phi}{\partial \pi_i} \to 0$, and the system converges to a Nash equilibrium.
> ***
>
> We also present the theoretical analysis of **system stability** of APC:
>
> ***
> Consider a homogeneous population of $N$ agents, all employing the APC algorithm. Each agent $i$ has an action space $a_i \in \{C, D\}$, and their strategy $\pi_i$ is updated via reinforcement learning to maximize expected return. The environment is an iterative public goods game with parameters $[n,e,r]$, satisfying $1 < r < n$.
>
> The total payoff for agent $i$ at time $t$ is:
> $$\Pi_i^{total}(t) = \Pi_i^{PGG}(t) - \sum_{j \neq i} w_{ji}(t)\delta - \sum_{j \neq i} w_{ij}(t)c$$
>
> where $\Pi_i^{PGG}(t)$ is the public goods game payoff, $w_{ji}(t)$ is the received punishment intensity, $w_{ij}(t)$ is the imposed punishment intensity, $\delta$ is the unit punishment harm, and $c$ is the unit punishment cost.
>
> Let $\rho_C(t)$ denote the proportion of cooperators, whose evolution satisfies:
>
> $$\dot{\rho} _ C(t) = \rho_D(t) W _ {D \to C}(t) - \rho_C(t) W _ {C \to D}(t)$$
>
> The transition probabilities are determined by the expected payoff difference:
>
> $$W_{D \to C} = \sigma(\mathbb{E}[\Pi_C] - \mathbb{E}[\Pi_D])$$
>
> $$W_{C \to D} = \sigma(\mathbb{E}[\Pi_D] - \mathbb{E}[\Pi_C])$$
>
> The expected payoffs for cooperators and defectors are respectively:
> $$\Pi_C^{total} = (\frac{\rho(n-1) e r}{n} - e+ \frac{e r}{n}) - (1-\rho)(n-1) c$$
>
> $$\Pi_D^{total} = \frac{\rho(n-1) e r}{n}- (1-\rho)(n-1) c -(n-1) \delta$$
>
> The payoff difference is:
>
> $$\Delta \Pi = \mathbb{E}[\Pi_C] - \mathbb{E}[\Pi_D] = -e+ \frac{e r}{n}+(n-1) \delta$$
>
> At the steady state $\dot{\rho}_C = 0$, substituting the transition probabilities yields:
>
> $$\rho_C^* = \sigma(\Delta \Pi)$$
>
> When $\Delta \Pi = 0$, the system is at a critical state, solving for the critical punishment threshold:
>
> $$\delta^* = \frac{e-\frac{e r}{n}}{(n-1)}$$
>
> When $\delta > \delta^* $ ,  $\Delta \Pi > 0$ leads to $W_{D \to C} > W_{C \to D}$, and the system tends toward cooperation; when $\delta < \delta^* $, the system tends toward defection; when $\delta = \delta^*$, the system stabilizes at a mixed equilibrium.
> ***

---

> ### Author Response · Authors · 2025-11-25
> **Response to Reviewer MTXx (4)**
>
> >Q5: The authors are also encouraged to evaluate APC in larger-scale social systems (e.g., with N > 10 agents) to assess its scalability and stability in more complex collective environments.
>
> We introduced a more complex, larger-scale heterogeneous Foraging environment involving 12 agents. The environment setup is based on the paper referenced as [1]. In Foraging, 12 agents move across a 2D grid and collect resources. Among them, 75% are common agents that can only collect common berries, while 25\% are special agents that can also collect forbidden berries (4 points). Initially, 70% of the resources in the environment are common berries, normally yielding a reward of 3 points. However, once a special agent collects a forbidden berry, it triggers permanent resource degradation, reducing the reward of all common berries to 1 point, and negatively affecting the rewards of all common agents. Each time a berry is collected, a new berry regenerates, with a 0.1 probability of being a forbidden berry and the rest being common berries. To enforce norms, the system includes a marking and punishment mechanism: agents that violate the norm by collecting forbidden berries are marked for 10 time steps. common agents can spend a cost of 0.7 points to punish other agent, forcing the latter to pay a fine of 0.7 points.
>
> We compared the APC method with other baselines, and the experimental results show that in this larger-scale heterogeneous environment, APC still achieves superior performance compared to the baselines. It effectively utilizes punishment to deter special agents from collecting forbidden berries, encouraging them to collect common berries instead, thereby enhancing collective returns.
>
> |Collective env reward/Collective reward|Foraging|
> |:-:|:-:|
> |**LOLA**|$191.630(\pm5.446)/ 191.621(\pm5.436)$|
> |**SI**|$193.783(\pm6.223)/ 193.234(\pm5.981)$|
> |**D3C**|$162.443(\pm10.634)/ 162.035(\pm10.146)$|
> |**LPOLA**|$60.645(\pm5.406)/ 60.645(\pm5.406)$|
> |**CNM**|$207.828(\pm0.846)/128.887(\pm12.176)$|
> |**APC**|$439.283(\pm3.373)/ 438.760(\pm3.455)$|
>
> In the table above, "Collective env reward" refers to collective environment rewards (which do not consider the impact of $c$ and $\delta$), as shown in Figures 3-4. On the other hand, "Collective reward" refers to rewards that consider the impact of $c$ and $\delta$, as shown in Figure 10 and Figure 13. The values in the table represent the averages after convergence over five random seeds, while the values in parentheses indicate the standard deviation.
>
> >Q6: The Method section would benefit significantly from including a pseudocode representation of the APC training and execution procedure. Although the textual description is detailed, the current structure is somewhat intricate; a structured algorithmic framework would make the methodology clearer, more concise, and easier to reproduce.
>
> We have included pseudocode for the APC training and execution process in the appendix on page 29.
>
> [1] Köster, R., Hadfield-Menell, D., Everett, R., Weidinger, L., Hadfield, G. K., & Leibo, J. Z. (2022). Spurious normativity enhances learning of compliance and enforcement behavior in artificial agents. Proceedings of the National Academy of Sciences (PNAS).
>
> Thank you for your valuable feedback. We look forward to your further suggestions and hope the revisions meet your expectations.

---

### Official Review · Reviewer_t21e · 2025-11-01

**Soundness:** 2
**Presentation:** 3
**Contribution:** 2
**Rating:** 2
**Confidence:** 3

**Summary:**

In multi-agent mixed-motive RL scenarios, the paper proposes to train a network to judge if other agents are defecting (defined based on whether the actions caused damage to other agents' rewards), and places punishment on such actions. Results show in 4 grid-based tasks, the method increases collaborative rewards.

**Strengths:**

1. Results show that the proposed method increases collective reward in mix-motive scenarios.
1. Interesting and reasonable ablation study on the effect of hyperparameters.
1. The definition of defection gets rid of access to other agent's internal information (such as their rewards, intentions), which makes sense in real-life domains where such information might be private.

**Weaknesses:**

1. The definition of defection, "when the probability of ajt exceeds the mean, it indicates that agent jtends to favor actions unfavorable to the focal agent i’s payoff", seems a bit arbitrary, and might over-punish any action to rationally optimize self rewards.  To answer this question, can you analyze mathematically how this definition rigorously distinguishes various mixed-motive cases, such as "harm others to benefit oneself" and "benefit oneself while also benefiting others, but at a smaller scale compared to the benefit to oneself"? Similarly, how it treats "harm others and harm oneselves as well, in cases where a harm is inevitable"? Or alternatively, do you have a mathematical description of what this definition equivalently captures so we can look at the expression and understand its mechanism, instead of the intuitive and subjective descriptions in lines 131-137?
1. The evaluation is only compared to 3 baselines by 2017. I wonder why various methods introduced in the related work section are not compared to. In specific, the authors mention "However, experimental results, shown as in Figure 3, have shown that merely relying on such punishment action combined with standard Independent MARL methods often fails to promote cooperation in SSDs.". But these methods are not directly compared to, which makes the statement in lack of support.

**Questions:**

1. grammar: line 131: "µi outputs predict probability distribution ..."
1. It would be clearer to compare the various definitions of defections in related work or preliminaries, and how you chose yours. This is a core concept in the motivation, but only explained subjectively.
1. Any experimental validation on how accurately the defection awareness network correlates with other agents' defection intentions? This also helps clarify my previous questions on what the network really measures.

---

> ### Author Response · Authors · 2025-11-25
> **Response to Reviewer t21e (1)**
>
> Thank you very much for your detailed feedback. In the following, we provide detailed responses to the question you raised.
> >W1: The definition of defection, "when the probability of ajt exceeds the mean, it indicates that agent jtends to favor actions unfavorable to the focal agent i’s payoff", seems a bit arbitrary, and might over-punish any action to rationally optimize self rewards. To answer this question, can you analyze mathematically how this definition rigorously distinguishes various mixed-motive cases, such as "harm others to benefit oneself" and "benefit oneself while also benefiting others, but at a smaller scale compared to the benefit to oneself"? Similarly, how it treats "harm others and harm oneselves as well, in cases where a harm is inevitable"? Or alternatively, do you have a mathematical description of what this definition equivalently captures so we can look at the expression and understand its mechanism, instead of the intuitive and subjective descriptions in lines 131-137?
>
> ***
> We conducted a theoretical derivation proving that:
>
> $$
> \sigma^{ij}(a^j) > \frac{1}{m} \quad \Rightarrow \quad U^i(a^j) < \bar{U}^i
> $$
>
> Here, the action space size is $m = |\mathcal{A}^j|$, the average payoff is $\bar{U}^i = \frac{1}{m}\sum U^i(a^{j'})$, and $U^i(a^j) = r^i(o^i, a^j, a^{-j})$. That is, when the defection probability of an action is higher than the uniform probability, the payoff that this action brings to agent $i$ is lower than the average payoff caused by other actions to agent $i$. **In other words, we link the defection probability to the agent's payoff: defection can be understood as this action being unfavorable for agent $i$'s payoff relative to the average payoff from other actions for agent $i$.** In the following analysis, we will fix the observation $o^i$ and the actions of other agents $a^{-j}$.
>
> Agent $i$ needs to estimate the defection probability $\sigma^{ij}(a^j)$ that agent $j$ takes action $a^j$, where $U^i(a^j)$ represents the payoff received by $i$. The training objective of defection prediction network $\mu^i$ is to maximize:
>
> $$J(\mu^i) = \mathbb{E}_{a^j \sim \sigma^{ij}}[-U^i(a^j)] + \beta H(\sigma^{ij})$$
>
> The first term encourages assigning high probability to low-payoff actions, and the second term is an entropy regularization term with weight $\beta > 0$, where $H(\sigma^{ij})$ is the Shannon entropy.
>
> By solving the constrained optimization problem using the Lagrangian method, we obtain the closed-form solution:
>
> $$\sigma^{ij}(a^j) = \frac{\exp(-U^i(a^j)/\beta)}{\sum_{a^{j'}}\exp(-U^i(a^{j'})/\beta)}$$
>
> From the condition $\sigma^{ij}(a^j) > 1/m$, we obtain $\exp(-U^i(a^j)/\beta) > \bar{E}$, where $\bar{E} = \frac{1}{m}{\sum_{a^{j'}}} \exp(-U^i(a^{j'})/\beta)$.
>
> Since $f(x) = \exp(-x/\beta)$ has a positive second derivative and is strictly convex, by Jensen's inequality:
>
> $$\bar{E} = \frac{1}{m} {\sum_{a^{j'}}} f(U^i(a^{j'})) > f\left( \frac{1}{m} {\sum_{a^{j'}}} U^i(a^{j'}) \right) = \exp(-\bar{U}^i/\beta)$$
>
> $$\Rightarrow  \exp(-U^i(a^j)/\beta) > \bar{E} > \exp(-\bar{U}^i/\beta)$$
>
> Since $f(x) = \exp(-x/\beta)$ is a monotonically decreasing function, it follows that:
> $$\Rightarrow  U^i(a^j) < \bar{U}^i$$
> ***
>
> Through the above derivation, defection here is defined as an action by the other party that brings me a lower payoff than the average payoff of their other possible actions. Therefore, the APC focuses solely on whether its own payoff is compromised, without concerning itself with changes in the payoffs of others. Without needing to access the rewards of other agents or identify the specific mixed-motive scenario, it can effectively detect defection actions.

---

> ### Author Response · Authors · 2025-11-25
> **Response to Reviewer t21e (2)**
>
> > W2: The evaluation is only compared to 3 baselines by 2017. I wonder why various methods introduced in the related work section are not compared to. In specific, the authors mention "However, experimental results, shown as in Figure 3, have shown that merely relying on such punishment action combined with standard Independent MARL methods often fails to promote cooperation in SSDs.". But these methods are not directly compared to, which makes the statement in lack of support.
>
> In our four previously mentioned experimental environments, a punitive action was incorporated into the action spaces of all agents to equip them with punishment capability. The performance was then tested, and the results indicate that for IA2C, merely adding the punitive action was insufficient—its performance remained consistently poor, virtually unchanged from its performance without the punitive action shown in Figures 3-4, Figures 10-13 and Figures 16-17.
>
> |Collective env reward/Collective reward|IPGG|Coingame|SSG|SSH|
> |:-:|:-:|:-:|:-:|:-:|
> |**IA2C**|$0.0019(\pm0.0002)/ 0.0015(\pm0.00007)$|$0.236(\pm0.007)/ 0.225(\pm0.002)$|$48.903(\pm0.372)/ 48.520(\pm0.366)$|$3.895(\pm0.017)/ 3.883(\pm0.016)$|
> |**IA2C_NP**|$0.0021(\pm0.0003)/ 0.0021(\pm0.0003)$|$0.232(\pm0.010)/ 0.232(\pm0.010)$|$49.216(\pm0.430)/ 49.216(\pm0.430)$|$3.912(\pm0.023)/ 3.912(\pm0.023)$|
>
> In the above table, "Collective env reward" refers to collective environment rewards (not considering the impact of $c$ and $\delta$), which can be found in Figures 3-4, Figure 11 and Figure 16. On the other hand, "Collective reward" refers to rewards that consider the impact of $c$ and $\delta$, which can be found in Figure 10, Figures 12-13 and Figure 17. The values in the table represent the averages after convergence over five random seeds, while the values in parentheses indicate the standard deviation. IA2C refers to a setting where a punitive action was incorporated into the action spaces of all agents to equip them with punishment capability, whereas IA2C_NP refers to the setting without the punitive action.
>
> We introduc five additional baselines for comparison: LOLA, SI, D3C, LPOLA, and CNM (refer to [1-5]). The experimental results are presented below:
>
> |Collective env reward/Collective reward|IPGG|Coingame|SSG|SSH|
> |:-:|:-:|:-:|:-:|:-:|
> |**LOLA**|$0.009(\pm0.008)/ 0.003(\pm0.002)$|$6.050(\pm0.917)/ 6.035(\pm0.916)$|$88.316(\pm5.521)/ 86.934(\pm6.5347)$|$3.900(\pm0.023)/ 3.903(\pm0.021)$|
> |**SI**|$0.00047(\pm0.00006)/ 0.00003(\pm0.00001)$|$2.673(\pm0.423)/ 2.655(\pm0.411)$|$109.291(\pm16.230)/ 109.101(\pm16.130)$|$3.910(\pm0.024)/ 3.905(\pm0.019)$|
> |**D3C**|$9.971(\pm0.004)/ 9.603(\pm0.167)$|$23.630(\pm0.511)/ 23.617(\pm0.512)$|$119.587(\pm0.024)/ 119.442(\pm0.031)$|$10.986(\pm0.154)/ 10.985(\pm0.155)$|
> |**LPOLA**|$4.239(\pm1.492)/ 4.239(\pm1.492)$|$0.010(\pm0.006)/ 0.010(\pm0.006)$|$31.804(\pm1.424)/ 31.804(\pm1.424)$|$0.014(\pm0.019)/ 0.014(\pm0.019)$|
> |**CNM**|$0.0278(\pm0.027)/-10.335(\pm0.730)$|$0.007(\pm0.0009)/-215.13(\pm0.095)$|$47.826(\pm0.678)/46.369(\pm0.922)$|$3.811(\pm0.017)/3.808(\pm0.012)$|
> |**APC**|$9.998(\pm0.001)/ 9.991(\pm0.004)$|$24.628(\pm0.017)/ 24.617(\pm0.015)$|$119.618(\pm0.237)/ 119.458(\pm0.221)$|$18.960(\pm0.091)/ 18.959(\pm0.091)$|
>
> In the table above, "Collective env reward" refers to collective environment rewards (which do not consider the impact of $c$ and $\delta$), as shown in Figures 3-4. On the other hand, "Collective reward" refers to rewards that consider the impact of $c$ and $\delta$, as shown in Figure 10 and Figure 13. The values in the table represent the averages after convergence over five random seeds, while the values in parentheses indicate the standard deviation.
>
> In our experimental environments, all baseline methods were equipped with a punitive action within their action spaces to enable punishment capability. The results demonstrate that APC consistently outperforms all five baselines. Notably, it maintains superiority even when compared to methods [1-3] that leverage extra private information, and exhibits more effective punishment strategies than specialized punishment approaches like [4-5].
>
> Note: The following provides a general overview of these methods. LOLA considers the learning process of other agents when updating its own policy parameters. SI achieves coordination by rewarding agents for having causal influence over other agents’ actions. D3C guides self-interested agents toward collectively efficient cooperative equilibria by having them mix rewards and follow the gradient of an efficiency bound during learning. LPOLA simultaneously predicts environmental actions and determines which actions of other agents to penalize, applying penalties when predictions match reality. CNM fosters cooperation by establishing social norms, whose punishments are socially enforced based on group consensus.

---

> ### Author Response · Authors · 2025-11-25
> **Response to Reviewer t21e (3)**
>
> > Q1: grammar: line 131: "µi outputs predict probability distribution ..."
>
> Thank you for the correction. The manuscript has been amended accordingly at line 131.
>
> >Q2: It would be clearer to compare the various definitions of defections in related work or preliminaries, and how you chose yours. This is a core concept in the motivation, but only explained subjectively.
>
> In the literature [4-5], the LPOLA and CNM methods offer distinct approaches to punishment in multi-agent systems. LPOLA employs reinforcement learning to directly maximize the total environmental reward while predicting environmental actions and "potential defection actions by opponents." When a predicted defection action aligns with the observed action, LPOLA treats it as defection and imposes punishment, which yields a positive reward for the agent; otherwise, punishment results in a negative reward. On the other hand, in the CNM method, defection is defined as behavior that violates group social norms. A network is used to learn the likelihood of group members executing punishment actions in the current state, quantifying the "group identification probability." When this probability exceeds 0.5, carrying out punishment yields a positive reward; otherwise, it results in a negative reward.
>
> The core idea of both LPOLA and CNM methods is to encourage punishment by attaching pseudo-rewards. However, this design has a drawback: since punishment can generate direct positive rewards, agents may develop a tendency to "punish for the sake of punishing." Experimental results in Figures 3-4, Figure 10 and Figure 13, show that a significant amount of punitive behavior persists in the later stages of training, which interferes with learning behaviors aimed at obtaining environmental rewards and leads to low collective returns.
>
> In contrast, our method, APC, defines "defection" as a situation where the reward an agent receives from the opponent’s current action is lower than the average reward the agent would expect from other possible actions of the opponent. This definition does not require predicting specific intentions of the opponent, as in LPOLA, nor does it rely on an abstract and potentially inconsistent social norm as in CNM. Moreover, it does not depend on pseudo-rewards to incentivize punishment. The judgment is entirely based on the actual rewards obtained by the individual, offering decentralized and adaptive advantages. Experimental results confirm that APC can more effectively establish stable cooperation and achieve better performance in SSDs shown in Figures 3-4, Figure 10 and Figure 13.
>
>
> >Q3: Any experimental validation on how accurately the defection awareness network correlates with other agents' defection intentions? This also helps clarify my previous questions on what the network really measures.
>
> To evaluate the identification accuracy of the DPN, we conducted experiments in both IPGG and SSH environments. In IPGG, we define the complete lack of contribution as defection behavior. In SSH, defection is defined as the act of hunting hare when the other agent is hunting stag. We collected 50,000 trajectory data points using a random policy. These trajectories were fed into the pre-trained defection-aware network, which outputs a probability distribution over the target agent's actions based on the current observation and the other agent's action. An action is classified as defection when the probability assigned by the network exceeds that of a uniform distribution over the action space. By comparing the network's predictions with the ground-truth defection labels, we calculated the identification accuracy. Experimental results show that the defection-aware network achieved an accuracy of 99.983% in the IPGG environment and 98.975% in the SSH environment. This demonstrates that the defection-aware network can effectively identify defection behaviors across different environments, providing a reliable foundation for the subsequent punishment mechanism.

---

> ### Author Response · Authors · 2025-11-25
> **Response to Reviewer t21e (4)**
>
> [1] Foerster, J., Chen, R. Y., Al-Shedivat, M., Whiteson, S., Abbeel, P., & Mordatch, I. (2018). Learning with Opponent-Learning Awareness (LOLA). In Proceedings of the 17th International Conference on Autonomous Agents and Multiagent Systems (AAMAS).
>
> [2] Jaques, N., Lazaridou, A., Hughes, E., Gulcehre, Ç., Ortega, P. A., Strouse, D. J., Leibo, J. Z., & de Freitas, N. (2019). Social Influence as Intrinsic Motivation for Multi-Agent Deep Reinforcement Learning. In Proceedings of the 36th International Conference on Machine Learning (ICML).
>
> [3] Gemp, I., McKee, K. R., Everett, R., Duéñez-Guzmán, E., Bachrach, Y., Balduzzi, D., & Tacchetti, A. (2022). D3C: Reducing the Price of Anarchy in Multi-Agent Learning. In Proceedings of AAMAS 2022.
>
> [4] Schmid, K., Belzner, L., & Linnhoff-Popien, C. (2021). Learning to penalize other learning agents. In Artificial Life Conference Proceedings.
>
> [5] Yaman, A., Leibo, J. Z., Iacca, G., & Wan Lee, S. (2023). The emergence of division of labour through decentralized social sanctioning. Proceedings of the Royal Society B.
>
> Thanks again for your thorough review. We hope our responses adequately address your concerns, and we would be grateful for any further comments or suggestions you may have.

---

> > ### Comment · Reviewer_t21e · 2025-11-27
> > **Follow-on Response**
> >
> > Thanks for the efforts in making the paper more solid. Even though D3C show strong performance more or less similar to the proposed APC, I think the comparison make the paper better grounded. The response to W1 still follows the original intuition in the paper, but looks a bit better by being added to the paper.
> >
> > Neverthelss, I plan to increase my score. Thanks for the hard work

---

> > > ### Author Response · Authors · 2025-11-28
> > >
> > > We thank the reviewer for their positive feedback. We apologize for any confusion regarding the performance comparison between D3C and our method, APC. We would like to kindly clarify that while APC and D3C achieve comparable results in the IPGG, CG, and SSG environments, **APC demonstrates superior performance in SSH and the more complex and larger-scale Foraging environments. Thus, APC shows better performance compared to D3C**. The training curves of SSH and Foraging for the baselines are provided in Figure 4(c) and Figure 4(d).
> > >
> > > |Collective env reward/Collective reward|SSH|Foraging|
> > > |:-:|:-:|:-:|
> > > |**D3C**|$10.986(\pm0.154)/ 10.985(\pm0.155)$|$162.443(\pm10.634)/ 162.035(\pm10.146)$|
> > > |**APC**|$18.960(\pm0.091)/ 18.959(\pm0.091)$|$439.283(\pm3.373)/ 438.760(\pm3.455)$|
> > >
> > > Foraging is a more complex, larger-scale heterogeneous Foraging environment involving 12 agents. The environment setup is based on the paper referenced as [1]. In Foraging, 12 agents move across a 2D grid and collect resources. Among them, 75% are common agents that can only collect common berries, while 25\% are special agents that can also collect forbidden berries (4 points). Initially, 70% of the resources in the environment are common berries, normally yielding a reward of 3 points. However, once a special agent collects a forbidden berry, it triggers permanent resource degradation, reducing the reward of all common berries to 1 point, and negatively affecting the rewards of all common agents. Each time a berry is collected, a new berry regenerates, with a 0.1 probability of being a forbidden berry and the rest being common berries. To enforce norms, the system includes a marking and punishment mechanism: agents that violate the norm by collecting forbidden berries are marked for 10 time steps. common agents can spend a cost of 0.7 points to punish other agent, forcing the latter to pay a fine of 0.7 points.
> > >
> > > Thank you once again for your encouraging comments on our work. We would be glad to discuss further if you have any additional questions.
> > >
> > > [1] Köster, R., Hadfield-Menell, D., Everett, R., Weidinger, L., Hadfield, G. K., & Leibo, J. Z. (2022). Spurious normativity enhances learning of compliance and enforcement behavior in artificial agents. Proceedings of the National Academy of Sciences (PNAS).

---

### Author Response · Authors · 2025-12-01
**General Responses**

Dear Area Chair,

We sincerely thank all the reviewers for their insightful comments and constructive suggestions. **We have thoroughly addressed all the reviewers' comments and concerns**. Our detailed responses, along with the corresponding revisions made throughout the manuscript, are provided below. We have categorized them into two main groups: **empirical results** and **theoretical analysis**.

### **Emperical Results**

1. **Additional Baselines** (@Reviewer t21e W2, qEjW W1, duR2 W4), **Ablations** (@Reviewer t21e W2, MTXx Q3), and **Large-Scale Heterogeneous Agent Setups** (@Reviewer MTXx Q5, qEjW W1, duR2 W1): We have added five baselines (LOLA, SI, D3C, LPOLA, CNM) and confirmed APC's superior performance (Sections 4.1.3, 4.2). The introduction of extra punishment actions has not affected results (Appendix F.2 in line 1371-1377), nor has the choice of RL algorithm (Appendix F.2 in line 1381-1385). In the 12-agent heterogeneous Foraging environment, our method have again demonstrated superiority (Sections 4.1.1, 4.2).

2. **The interpretation of Defection Probability of DPN** (@Reviewer t21e W1) and **Comparison with Other Punishment-based Methods** (@Reviewer t21e Q2): Using Lagrangian approach and Jensen’s inequality, we have proven that an action with a higher-than-uniform defection probability provides a lower payoff to the agent than the average payoff from other actions (Appendix C). We have also compared the advantages of our defection definition over punishment-based methods (LPOLA and CNM) and provided experimental validation (Section 4.2 and Appendix F.1).

3. **Accuracy** (@Reviewer t21e Q3, duR2 Q2&W2) and **Generalization** (@Reviewer duR2 Q2) of **the DPN network**: We have experimentally validated that the DPN network achieves high accuracy (Appendix F.5, lines 1444–1479) in terms of both training accuracy and convergence accuracy, as well as its ability to generalize across different adversaries or environments (Appendix F.5, lines 1480–1485).

### **Theoretical Analysis**

1. **Second-order Free-rider Problem** (@Reviewer MTXx Q1): Using replicator dynamics, we have demonstrated that APC can avoid the second-order free-rider problem when the punishment intensity is sufficient (Appendix B.5).

2. **Convergence and System Stability of APC** (@Reviewer MTXx Q4, qEjW W2): Through Lyapunov stability theory and replicator dynamics, we have proven APC converges to Nash equilibrium and results in stable cooperation when punishment is sufficient.（Appendix B.4）

3. **Convergence of Punishment Probability** (@Reviewer duR2 Q1): From the perspectives of stochastic processes and dynamical systems, we have proven that the punishment probability update rule, Eq. (3), is guaranteed to converge (Appendix D).

4. **Generalization and Extension of Penalty Parameter Analysis** (@Reviewer duR2 Q4&W3): We proved the generalizability of the penalty parameter analysis across different environmental parameters (Appendix B.2); and have provided a theoretical extension and proof in stochastic sequential games (Appendix B.3).

**We have also revised the manuscript according to the reviewers' comments**, including corrections of typos and pictures presentations (@Reviewer t21e Q1, line 131; @Reviewer qEjW Q1&Q2, lines 385–392 and line 1175), revisions to the Related Work section (@Reviewer MTXx Q2, Section 5), supplement to the pseudocode representation (@Reviewer MTXx Q6, Page 29), and corrections to the fairness metrics (@Reviewer duR2 Q3, Section 4.2).
***
### **Reviewer updates**
After these revisions and clarifications:
- **Reviewer t21e** explicitly states that our revisions have made the paper more solid and better grounded, and **has raised their score to 6**.
- **Reviewer MTXx** indicates that if concerns have been solved, they will **raise their score**; we have also supplemented rich experiments and two theoretical derivations, and in the paper revised the related work and added pseudocode, addressing their concerns.
- **Reviewer qEjW** already had a positive score, and their main concerns—extensive baselines, experiments in large-scale heterogeneous environment, and theoretical derivations—have been addressed in our response.
- **Reviewer duR2**'s primary worries — including the interpretation of DPN's output, its performance, the sufficiency of baselines, and the generalizability of the theoretical derivation — also have been resolved.

In summary, the revision offers a clearer and theoretically stronger decision-making method to promote cooperation in mixed-motive games, with more interpretable and detailed analysis of DPN, new supporting experiments, and four additional theoretical contributions.

---

### Meta-Review · Area_Chair_oCGD · 2026-01-07

**Summary:**

This paper studies the problem of cooperation in MARL. In particular, it investigates the issue of defection and subsequent punishment to disincentivise defection.

The reviews were quite mixed, highlighting significant concerns. In particular, I believe that the concept of defection should be discussed much more. I am not convinced that, the way it is defined in this paper, would correctly characterise real-world defection.

For the reasons I detail below, I believe this paper is under the acceptance threshold.

**Reviewer Concerns:**

## Definition of defection

In the rebuttal, the authors define defection as:
>In contrast, our method, APC, defines "defection" as a situation where the reward an agent receives from the opponent’s current action is lower than the average reward the agent would expect from other possible actions of the opponent.

It seems to be a very contrived definition as:
- we can imagine a setting where an give very low reward to everyone making all the other actions above the average. More concretely, we could imagine a version or the prisoner's dilemma where there is a third action that gives a reward of -1000 for both agents (regardless of what the other does). In that case the payoff of defecting will always induce an utily above the average and thus would not be a defection according to your definition
- Your definition does not consider cumulative reward. For instance, if the negative reward due to the defection was delayed (which is the case in many practical social dilemmas) the action responsible for the delayed negative reward would not be considered as defection.

Moreover, I believe this second point is closely related to the reviewer's concerns about the scale of the experiments. I think that considering tasks where the reward due to defection is delayed (e.g. common harvest) would significantly improve the contribution of this work.

I encourage the authors to address these aspects in the subsequent revision.


## convergence proof for APC:

The "convergence proof for APC" that is provided in the appendix only works when $\frac{\partial E[\Pi^{PGG}]_i}{\partial \pi_j} = 0$. Otherwise, it would imply that simultaneous gradient ascent on each player's payoff would converge to the Nash equilibrium for any game (which we know is not true; take, for instance, matching pennies).

Moreover, the setting where $\frac{\partial E[\Pi^{PGG}]_i}{\partial \pi_j} = 0$ does not seem to be the setting of interest for cooperative games (we want to be in a setting where the action of other players influences the reward of player j)

However, I acknowledge that showing convergence of such an algorithm in the general MARL setting is out of scope for this paper (given that it is not even true for standard policy-gradient theorems).

An alternative theoretical result could be, in some specific cases, such as the one presented in the rebuttal.

**Reviewer Scores:**

I do not think they would have changed their score significantly to make this paper above the acceptance threshold.

---

### Decision · Program_Chairs · 2026-01-26

Reject